# Oligodendrocytes control potassium accumulation in white matter and seizure susceptibility

Valerie A Larson[1], Yevgeniya Mironova[1], Kimberly G Vanderpool[2], Ari Waisman[3], John E Rash[2], Amit Agarwal[1†*], Dwight E Bergles[1*]

[1]The Solomon H. Snyder Department of Neuroscience, Johns Hopkins University School of Medicine, Baltimore, United States; [2]Department of Biomedical Sciences, Colorado State University, Fort Collins, United States; [3]Institute for Molecular Medicine, University Medical Center of the Johannes Gutenberg University, Mainz, Germany

**Abstract** The inwardly rectifying $K^+$ channel $K_{ir}4.1$ is broadly expressed by CNS glia and deficits in $K_{ir}4.1$ lead to seizures and myelin vacuolization. However, the role of oligodendrocyte $K_{ir}4.1$ channels in controlling myelination and $K^+$ clearance in white matter has not been defined. Here, we show that selective deletion of $K_{ir}4.1$ from oligodendrocyte progenitors (OPCs) or mature oligodendrocytes did not impair their development or disrupt the structure of myelin. However, mice lacking oligodendrocyte $K_{ir}4.1$ channels exhibited profound functional impairments, including slower clearance of extracellular $K^+$ and delayed recovery of axons from repetitive stimulation in white matter, as well as spontaneous seizures, a lower seizure threshold, and activity-dependent motor deficits. These results indicate that $K_{ir}4.1$ channels in oligodendrocytes play an important role in extracellular $K^+$ homeostasis in white matter, and that selective loss of this channel from oligodendrocytes is sufficient to impair $K^+$ clearance and promote seizures.

DOI: https://doi.org/10.7554/eLife.34829.001

*For correspondence:
agarwal@ana.uni-heidelberg.de (AA);
dbergles@jhmi.edu (DEB)

Present address: †Institute for Anatomy and Cell Biology, Heidelberg University, Heidelberg, Germany

## Introduction

Action potentials lead to increases in extracellular $K^+$ that if left unchecked can impair membrane repolarization, induce tonic firing and trigger seizures. The redistribution of extracellular $K^+$ following neuronal activity is mediated through diffusion and uptake into cells ($K^+$ buffering), particularly glial cells which maintain a high resting conductance to $K^+$ and a highly negative resting potential (*Kofuji and Newman, 2004*). In gray matter, astrocytes have been shown to participate in $K^+$ buffering, but much less is known about the mechanisms that enable $K^+$ clearance in white matter, where astrocyte access to axons is limited to nodes of Ranvier and the majority of axonal $K^+$ channels are located beneath the myelin sheath (*Rash et al., 2016*; *Wang et al., 1993*). Oligodendrocytes have extended contact with axons and exhibit a high resting conductance to $K^+$, but their contribution to $K^+$ clearance in white matter and the impact of this $K^+$ redistribution on neuronal activity has not been determined.

$K^+$ entry into glial cells is facilitated, in part, by inwardly rectifying $K^+$ channels ($K_{ir}$ channels) that promote unidirectional movement of $K^+$ across membranes. $K_{ir}4.1$ is the most abundant $K^+$ channel expressed by astrocytes and oligodendroglia (*Zhang et al., 2014*) and has been shown to help establish their resting membrane potential (*Djukic et al., 2007*; *Neusch et al., 2006*) and to redistribute extracellular $K^+$ in gray matter following neuronal activity (*Chever et al., 2010*; *Haj-Yasein et al., 2011*; *Neusch et al., 2006*). The importance of $K_{ir}4.1$ in $K^+$ homeostasis is underscored by the severe neurological phenotype of $K_{ir}4.1$ knockout mice, which exhibit ataxia, seizures,

deafness, widespread myelin pathology, and early death (*Djukic et al., 2007*; *Neusch et al., 2001*). Many of these features are also observed in human patients with SeSAME/EAST syndrome, a rare genetic disease caused by loss-of-function mutations in $K_{ir}4.1$ (*Bockenhauer et al., 2009*; *Scholl et al., 2009*). Polymorphisms in the $K_{ir}4.1$ gene are associated with idiopathic epilepsy (*Buono et al., 2004*; *Heuser et al., 2010*; *Lenzen et al., 2005*) and autism spectrum disorder with seizures (*Sicca et al., 2011*; *2016*), futher supporting a close functional link between Kir4.1 dysfunction and neuronal hyperexcitability. $K_{ir}4.1$ expression levels are reduced in a vast array of CNS pathologies (for review, see *Nwaobi et al., 2016*), and restoration of astrocyte $K_{ir}4.1$ levels in the striatum of Huntington's disease model mice enhanced neuronal suvival and ameliorated motor deficits (*Tong et al., 2014*), suggesting that impaired $K^+$ buffering may contribute to both genetic and aquired neurological disease.

Although the pathological sequelae arising from $K_{ir}4.1$ dysfunction are commonly attributed to astrocytes, oligodendrocyte precursor cells (OPCs) and mature oligodendrocytes have also been shown to express $K_{ir}4.1$, by immunohistochemistry (*Kalsi et al., 2004*; *Poopalasundaram et al., 2000*), reporter gene expression (*Tang et al., 2009*), RNA-Seq (*Tasic et al., 2016*; *Zhang et al., 2014*) and recordings of $K_{ir}$-mediated currents (*Battefeld et al., 2016*; *Maldonado et al., 2013*). OPCs that lack $K_{ir}4.1$ fail to fully mature into oligodendrocytes in vitro (*Neusch et al., 2001*), and white matter tracts in glial-specific $K_{ir}4.1$ knockout mice exhibit extensive vacuolization (*Djukic et al., 2007*), suggesting that this $K^+$ channel performs diverse roles in oligodendroglia. Functional studies indicate that selective depolarization of oligodendrocytes in white matter alters the conduction speed of action potentials, an effect that is abolished by application of a $K_{ir}$ channel blocker (*Yamazaki et al., 2007*; *2014*), suggesting that oligodendrocyte $K_{ir}4.1$ channels can shape neuronal activity on a rapid time scale. However, the relative contribution of oligodendroglial $K_{ir}4.1$ has been difficult to assess in vivo, as pharmacological manipulations and genetic deletion studies, performed in either global knockouts (*Neusch et al., 2001*) or early progenitor cells that give rise to both astrocytes and oligodendroglia (*Djukic et al., 2007*), do not distinguish the contributions of these two cell types. As a result, the role of $K_{ir}4.1$ channels in regulating oligodendrocyte development and function remains poorly understood.

Here, we used conditional genetic strategies to define the role of $K_{ir}4.1$ channels in shaping oligodendrocyte development and maintaining myelin, and the participation of oligodendrocytes in controlling $K^+$ clearance and neuronal activity within white matter. Although selective deletion of $K_{ir}4.1$ from OPCs raised their membrane potential and membrane resistance, it did not alter their survival, proliferation or their ability to form oligodendrocytes in vivo. Selective deletion of this channel from mature oligodendrocytes was similarly benign, as it had no effect on their survival or ability to form myelin. However, despite the appearance of normal myelin in the absence of oligodendrocyte $K_{ir}4.1$, these mice exhibited spontaneous seizures, increased chemoconvulsant sensitivity, and activity-dependent deficits in motor behavior. Physiological studies in the corpus callosum and optic nerve revealed that $K^+$ clearance rates in white matter of these mice were markedly slowed and axons were impaired in their ability to sustain repetitive firing. Together, these findings indicate that oligodendrocytes play a crucial role in controlling $K^+$ homeostasis and that disruption of $K_{ir}4.1$ expression in oligodendrocytes is sufficient to promote neuronal hyperexcitability and seizures.

## Results

### CNS deletion of $K_{ir}4.1$ alters the structure but not the development of oligodendrocytes

Constitutive deletion of $K_{ir}4.1$ from glia (G*fa2-Cre;Kcnj10*$^{fl/fl}$ mice) leads to dramatic changes in the structure of myelin and in vitro studies of oligodendrocytes lacking $K_{ir}4.1$ suggest that this channel is required for both their maturation and survival (*Djukic et al., 2007*; *Neusch et al., 2001*). However, the role of $K_{ir}4.1$ in shaping the development of oligodendroglia in vivo remains largely unexplored. We deleted $K_{ir}4.1$ from the nervous system by crossing *Kcnj10*$^{fl/fl}$ mice (*Djukic et al., 2007*) with *Nes-Cre* mice, which express Cre recombinase in neural progenitor cells (*Tronche et al., 1999*). These nervous-system-specific $K_{ir}4.1$ conditional knockout mice (termed $nK_{ir}4.1cKO$) exhibited retarded growth, ataxia, tremor, and early mortality, with most mice dying by postnatal day (P) 25 (*Figure 1A–B*; *Figure 1—video 1*), similar to the phenotype of $K_{ir}4.1$ global knockout and glia-

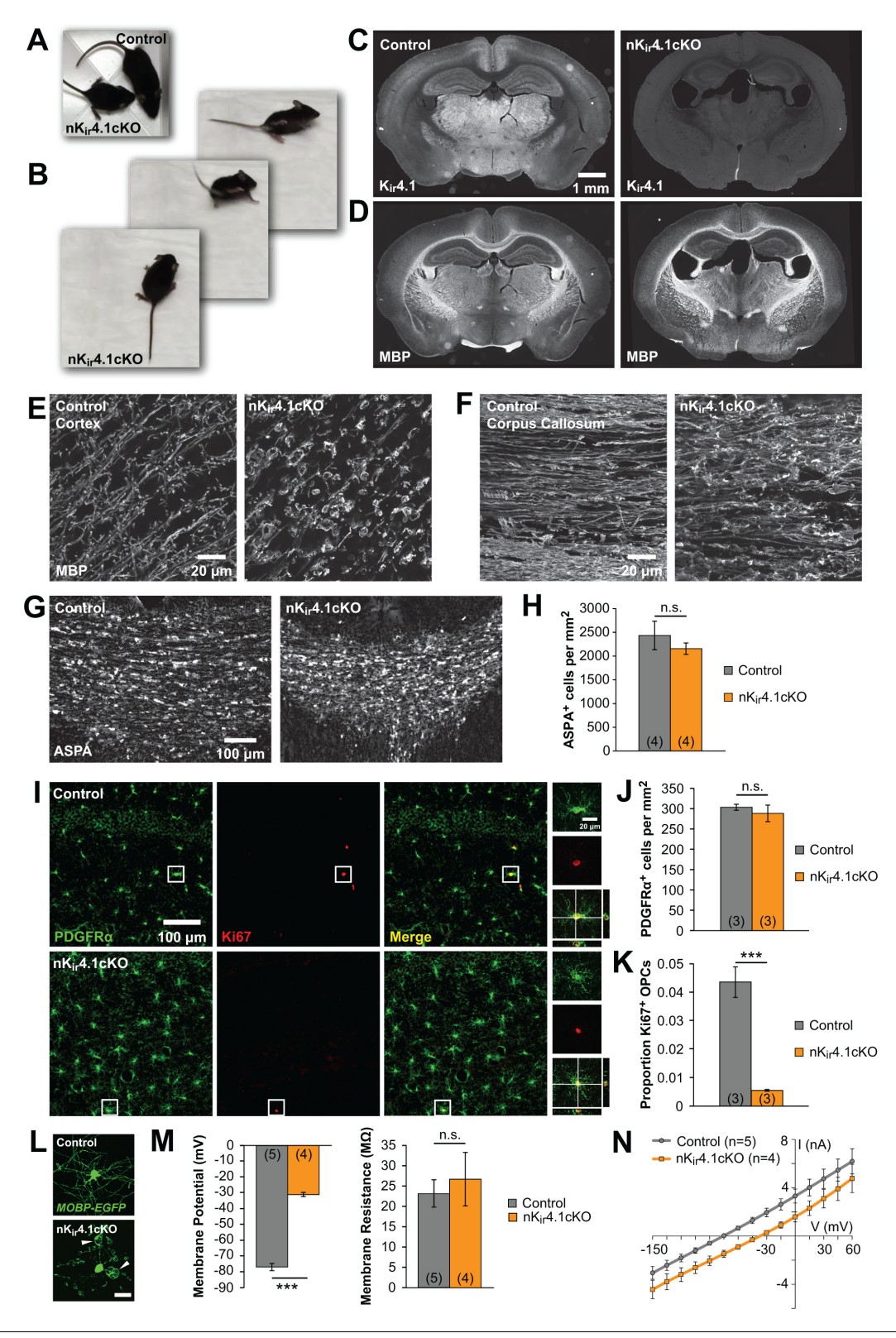

**Figure 1.** The fate of oligodendroglia following constitutive deletion of Kir4.1 from the CNS. (**A**) A P24 *Nes-Cre;Kcnj10^{fl/fl}* (nKir4.1cKO) mouse and a *Nes-Cre;Kcnj10^{fl/+}* (control) littermate. (**B**) nKir4.1cKO mice display ataxic gate (see *Figure 1—video 1*). (**C**) Coronal brain sections from control (left) and nKir4.1cKO (right) mice at P24, immunostained with antibody against Kir4.1. (**D**) Coronal brain sections from control (left) and nKir4.1cKO (right) mice at P24, immunostained with antibody against Myelin Basic Protein (MBP). (**E–F**) Higher magnification images of cortex (**E**) and corpus callosum (**F**) in

*Figure 1 continued on next page*

*Figure 1 continued*

control and nK$_{ir}$4.1cKO mice, immunostained for MBP. Images of additional brain regions are included in *Figure 1—figure supplement 1*. (**G**) Immunostaining for aspartoacylase (ASPA, a marker of mature oligodendrocytes) in corpus callosum of control (left) and nK$_{ir}$4.1cKO (right) mice. (**H**) Quantification of ASPA$^+$ cells per mm$^2$ in control ($n$ = 4) and nK$_{ir}$4.1cKO ($n$ = 4) mice. No significant difference in oligodendrocyte density was observed (p=0.60; Student's t-test). (**I**) Immunostaining for PDGFRα (green) and Ki67 (red) in hippocampus of control (top row) and nK$_{ir}$4.1cKO (bottom row) mice. Insets show PDGFRα and Ki67 double-immunoreactive cells. (**J**) Quantification of PDGFRα$^+$ cells per mm$^2$ in hippocampus of control ($n$ = 3) and nK$_{ir}$4.1cKO ($n$ = 3) mice. No significant difference in OPC density was observed (p=0.52; Student's t-test). (**K**) Quantification of the proportion of PDGFRα$^+$ cells that were Ki67$^+$ in hippocampus of control ($n$ = 3) and nK$_{ir}$4.1cKO ($n$ = 3) mice. Significantly fewer OPCs were Ki67$^+$ in nK$_{ir}$4.1cKO mice (p=0.002; Student's t-test). (**L**) Layer 1 oligodendrocytes in cortex (flat-mount) from *Mobp-EGFP* (control, top) and nK$_{ir}$4.1cKO;*Mobp-EGFP* (bottom) mice. Scale bar = 20 µm. White arrowheads indicate large myelin vacuoles. (**M**) Resting membrane potential and membrane resistance of corpus callosum oligodendrocytes recorded in acute slices from control (*Nes-Cre;Mobp-EGFP*, $n$ = 5 cells) and nK$_{ir}$4.1cKO (*Nes-Cre;Kcnj10$^{fl/fl}$;Mobp-EGFP*, $n$ = 4 cells) mice at P21. nK$_{ir}$4.1cKO oligodendrocytes had a depolarized membrane potential compared to controls (p=7.6 × 10$^{-7}$; Student's t-test), but no significant difference was observed in membrane resistance (p=0.62; Student's t-test). (**N**) I-V curves of control (gray) and nK$_{ir}$4.1cKO (orange) corpus callosum oligodendrocytes. nK$_{ir}$4.1cKO cells were depolarized, but maintained a linear I-V curve.

DOI: https://doi.org/10.7554/eLife.34829.002

The following video and figure supplement are available for figure 1:

**Figure supplement 1.** Immunostaining for MBP in layer 1 of cortex (flat mount preparation) (**A**), striatum (**B**), cerebellum (**C**), and spinal cord dorsal column (**D**) of control and nK$_{ir}$4.1cKO mice.
DOI: https://doi.org/10.7554/eLife.34829.003

**Figure 1—video 1.** *Nes-Cre;Kcnj10$^{fl/fl}$* (nK$_{ir}$4.1cKO) mouse at P24 displays an ataxic gait.
DOI: https://doi.org/10.7554/eLife.34829.004

specific K$_{ir}$4.1 knockout mice (*Djukic et al., 2007*; *Neusch et al., 2001*). K$_{ir}$4.1 immunoreactivity was no longer observed in the brains of these mice (*Figure 1C*), demonstrating effective deletion of this channel from the CNS. Immunostaining for myelin basic protein (MBP) revealed dense myelinated tracts in the corpus callosum and striatum, indicating that oligodendrocytes were formed; however, these regions exhibited widespread vacuolization of myelin, with swellings visible along most internodes (*Figure 1D–F*; *Figure 1—figure supplement 1*). Despite these striking morphological changes, the density of oligodendrocytes in nK$_{ir}$4.1cKO mice at P24, measured by immunostaining for the oligodendrocyte protein aspartoacylase (ASPA), was comparable to control mice (*Nes-Cre;Kcnj10$^{fl/+}$*) (*Figure 1G,H*), indicating that oligodendrocytes are generated and mature in the absence of K$_{ir}$4.1.

Oligodendrocytes are maintained by resident OPCs, which proliferate to replace those that transform into new oligodendrocytes (*Hughes et al., 2013*). Because of this robust homeostatic response, oligodendrocyte density can be maintained despite profound loss (*Kang et al., 2010*), which could mask underlying oligodendrocyte death. To assess whether there is accelerated turnover of oligodendrocytes in nK$_{ir}$4.1cKO mice, we examined the proportion of OPCs that were immunoreactive to the cell division marker Ki67 (*Figure 1I*). This analysis revealed that OPC density was unchanged (*Figure 1J*) in nK$_{ir}$4.1cKO mice, and their proliferation was reduced, rather than increased (*Figure 1K*), suggesting that oligodendrocyte turnover is not increased.

To assess the physiological properties of oligodendrocytes when K$_{ir}$4.1 is absent from the CNS, we crossed nK$_{ir}$4.1cKO mice with *Mobp-EGFP* mice (*Gong et al., 2003*), allowing oligodendrocytes to be visualized for targeted recording in acute slices (*Figure 1L*). EGFP$^+$ oligodendrocytes in the corpus callosum of these mice at P21 were significantly depolarized compared to controls (*Nes-Cre; Mobp-EGFP*) (control: –77 ± 2 mV, $n$ = 5; cKO: –31 ± 1 mV, $n$ = 4; p=7.6 × 10$^{-7}$) (*Figure 1M*), but maintained the membrane resistance and linear current-voltage response characteristic of WT oligodendrocytes (*Figure 1M,N*). Together these results indicate that deletion of K$_{ir}$4.1 from the CNS leads to profound changes in the structure of myelin and depolarization of oligodendrocytes but does not impair their maturation or accelerate their degeneration during development.

## K$_{ir}$4.1 channels contribute to the resting membrane potential and membrane resistance of OPCs

K$_{ir}$4.1 is the most abundant K$^+$ channel expressed by OPCs (*Larson et al., 2016*), and upregulation of these channels, as assessed by the appearance of Ba$^{2+}$-sensitive K$_{ir}$ currents, correlates with development of a more negative OPC membrane potential and acquisition of a linear I-V curve profile over the course of postnatal development (*Maldonado et al., 2013*), suggesting that K$_{ir}$4.1 plays

a key role in establishing the membrane properties of these progenitors. To determine if $K_{ir}4.1$ channels influence the physiological properties of OPCs and their developmental progression in vivo, we selectively deleted these channels in OPCs by crossing *Pdgfra-CreER* mice (*Kang et al., 2010*) with *Kcnj10^{fl/fl}* mice. A sensitive Cre-dependent EGFP reporter transgene (*ROSA26-CAG-EGFP* (RCE)) (*Sousa et al., 2009*) was included to mark cells in which Cre activity was induced and recombination was initiated at P21 with 4-hydroxytamoxifen (4-HT). EGFP$^+$ OPCs in the corpus callosum, hippocampus, and cortex of these conditional knock-out mice (termed pK$_{ir}$4.1cKO) were targeted for whole cell recording in acute slices and compared to OPCs in control mice (*Pdgfra-CreER*;RCE) (*Figure 2A*). OPCs were readily distinguished from oligodendrocytes generated in the intervening 2 weeks by their radial morphology (*Figure 2B*) and presence of a depolarization-induced sodium current (*Figure 2C*) (*Bergles et al., 2000*; *De Biase et al., 2010*; *Ong and Levine, 1999*). OPCs in control mice exhibited prominent outward and inward currents in response to current injection, resulting in a near linear current-voltage (I-V) relationship (*Figure 2D,F*; *Figure 2—figure supplement 1A*). Application of a low concentration of BaCl$_2$ (100 μM), which inhibits $K_{ir}4.1$ channels, blocked the hyperpolarization-induced inward current (*Figure 2D,F*; *Figure 2—figure supplement 1A,B*). In contrast, the majority of OPCs (49/65, 76%) in pK$_{ir}$4.1cKO mice exhibited greatly reduced inward currents in response to hyperpolarization, and BaCl$_2$ (100 μM) did not alter their I-V relationship, indicating that $K_{ir}4.1$ was no longer expressed (*Figure 2E,G*; *Figure 2—figure supplement 2B*). The remaining 24% of EGFP$^+$ OPCs in pK$_{ir}$4.1cKO mice exhibited a linear I-V profile (*Figure 2—figure supplement 2*), suggesting that activation of CreER in these cells did not result in excision of both floxed $K_{ir}4.1$ alleles over this time period.

OPCs in both gray and white matter that lacked $K_{ir}4.1$ were ~20 mV more depolarized and exhibited a ~ 10 fold higher membrane resistance (measured at –80 mV with a 10 mV depolarization step) than control OPCs (*Figure 2H–I*; *Figure 2—figure supplement 1C–D*). Similar changes were induced in OPCs of control mice when $K_{ir}4.1$ was inhibited with 100 μM BaCl$_2$ (*Figure 2H–I*; *Figure 2—figure supplement 1C–D*), indicating that these changes reflect the direct contribution of $K_{ir}4.1$ rather than other compensatory changes induced by deletion of these channels. These results indicate that $K_{ir}4.1$ channels dominate the resting conductance of OPCs and force these progenitors to adopt a more hyperpolarized membrane potential.

## Deletion of $K_{ir}4.1$ from OPCs does not affect their survival, proliferation, or differentiation

Previous in vitro studies have shown that expression of $K_{ir}4.1$ is sufficient to hyperpolarize and cause $G_1/G_0$ arrest of glioma cells (*Higashimori and Sontheimer, 2007*) and flux of K$^+$ through voltage-gated K$^+$ channels has been shown regulate OPC proliferation and differentiation in vitro (*Gallo et al., 1996*; *Ghiani et al., 1999*; *Knutson et al., 1997*). To determine how the depolarization induced by $K_{ir}4.1$ deletion influences OPC behavior, we performed in vivo fate tracing using pK$_{ir}$4.1cKO mice bred to *R26R-EYFP* mice (*Srinivas et al., 2001*). In the RCE reporter line used for electrophysiological recordings, nearly 100% of OPCs expressed EGFP (*Figure 3—figure supplement 1*), yet $K_{ir}4.1$ currents were absent in only 76% of OPCs (*Figure 2—figure supplement 2*). Therefore, for these studies we used a *R26R-EYFP* reporter line that exhibits a recombination efficiency (78% of OPCs) comparable to the proportion of OPCs that lack $K_{ir}4.1$ currents in cKO mice (*Figure 3—figure supplement 1*). Control (*n* = 6) and pK$_{ir}$4.1cKO (*n* = 7) mice received 4-HT at P21 and the density of OPCs in gray and white matter was assessed at P35 by immunostaining for PDGFRα to identify all OPCs, and EYFP to identify OPCs that have expressed Cre recombinase and lack $K_{ir}4.1$. If OPC proliferation or survival is impaired, a decrease in the relative proportion of EYFP$^+$ OPCs is expected; however, there was no change in the density of OPCs or the proportion of EYFP$^+$ OPCs between control and pK$_{ir}$4.1cKO mice (*Figure 3—figure supplement 2*).

To assess proliferation more directly, the proportion of dividing OPCs was determined by administering bromodeoxyuridine (BrdU) twice daily from P28 to P34 (*Figure 3A*). Although the proportion of BrdU$^+$ OPCs varied by region, from ~45% in hippocampus to ~60% in corpus callosum, no differences were observed between EYFP$^+$ and EYFP$^-$ populations in either region (*Figure 3B–E*; *Figure 3—figure supplement 3A–C*), indicating comparable rates of proliferation. At this time point, EYFP$^+$ cells consisted of ~75% PDGFRα$^+$ OPCs and ~15% CC1$^+$ oligodendrocytes in the hippocampus, and ~45% OPCs and ~40% oligodendrocytes in the corpus callosum, with no significant differences observed between control and $K_{ir}4.1$-deleted OPCs (*Figure 3F–H*; *Figure 3—figure*

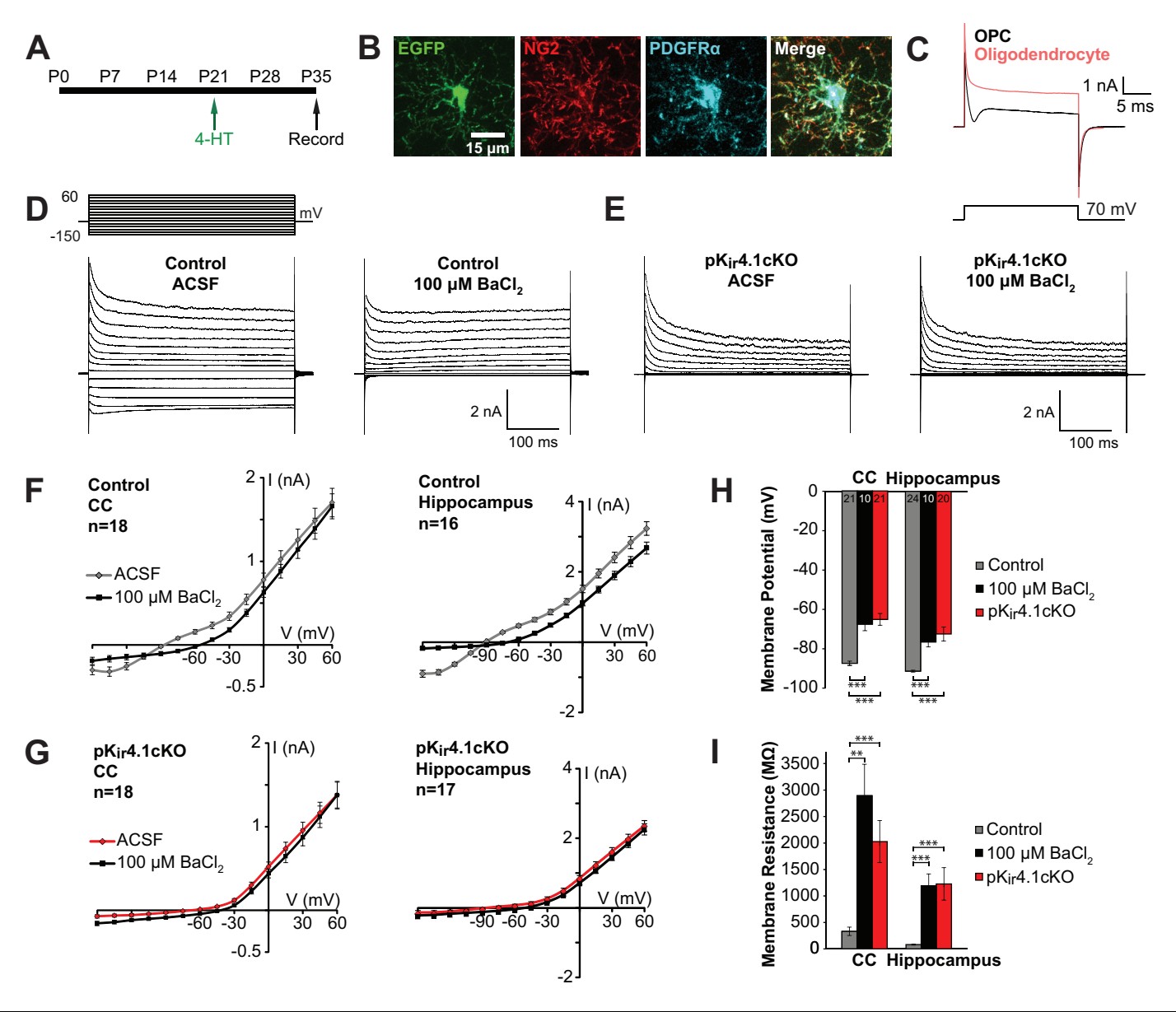

**Figure 2.** Effect of specific $K_{ir}4.1$ deletion on the membrane properties of OPCs. (**A**) Experimental protocol: 4-hydroxytamoxifen ($2 \times 1$ mg) was administered to *Pdgfra-CreER*;RCE (control) and *Pdgfra-CreER*;RCE;*Kcnj10^{fl/fl}* (pK$_{ir}$4.1cKO) mice i.p. at P21, and mice were sacrificed at P35 for whole cell recording in acute brain slices. (**B**) Immunostaining in fixed tissue from control mouse showing that EGFP (green) is expressed in OPCs, immunoreactive to NG2 (red) and PDGFRα (blue). (**C**) Response of an OPC (black) and an oligodendrocyte (red) to a 70 mV depolarizing step. The OPC shows a distinct inward deflection after the initial capacitive current, reflecting activation of voltage-gated sodium channels that are only expressed at the OPC stage. (**D**) Whole cell recordings from hippocampal OPCs of control mice. Voltage steps ($-150$ mV to $+60$ mV, 15 mV intervals) were applied to each cell in regular ACSF and in ACSF $+100\ \mu M$ $BaCl_2$. (**E**) The same voltage protocol was applied to hippocampal OPCs in pK$_{ir}$4.1cKO mice. (**F**) Plots showing the I-V relationship of control OPCs in corpus callosum (left, $n = 18$) and hippocampus (right, $n = 16$), in regular ACSF (gray) and ACSF $+100\ \mu M$ $BaCl_2$ (black). (**G**) Plots showing the I-V relationship of pK$_{ir}$4.1cKO OPCs in corpus callosum (left, $n = 18$) and hippocampus (right, $n = 17$), in regular ACSF (red) and ACSF $+100\ \mu M$ $BaCl_2$ (black). I-V plots for cortical OPCs are included in *Figure 2—figure supplement 1*. (**H**) Quantification of the average resting membrane potential of control OPCs in ACSF (gray, CC: $n = 21$, Hippocampus: $n = 24$), control OPCs in $100\ \mu M$ $BaCl_2$ (black, CC: $n = 10$, Hippocampus: $n = 10$), and pK$_{ir}$4.1cKO OPCs (red, CC: $n = 21$, Hippocampus: $n = 20$). The three groups are significantly different (CC: $F = 27.3$, $p=1.1 \times 10^{-8}$; Hippocampus: $F = 19.5$, $p=5.1 \times 10^{-7}$) (one-way ANOVA). Differences between control/ACSF and control/$BaCl_2$ (CC: $p=1.3 \times 10^{-4}$, Hippocampus: $p=1.9 \times 10^{-4}$) and between control/ACSF and pK$_{ir}$4.1cKO (CC: $p=2.4 \times 10^{-7}$, Hippocampus: $p=4.0 \times 10^{-5}$) are statistically significant (Bonferroni's test, $\alpha = 0.017$). CC = corpus callosum. (**I**) Quantification of membrane resistance of OPCs in (**H**). The three groups are significantly different (CC: $F = 13.2$, $p=2.7 \times 10^{-5}$; Hippocampus: $F = 10.8$, $p=1.2 \times 10^{-4}$) (one-way ANOVA). Differences between control/ACSF and control/$BaCl_2$ (CC: $p=0.0019$, Hippocampus: $p=7.7 \times 10^{-4}$) and between control/ACSF and pK$_{ir}$4.1cKO (CC: $p=3.9 \times 10^{-4}$, Hippocampus: $p=0.0014$) are

*Figure 2 continued on next page*

*Figure 2 continued*

statistically significant (Bonferroni's test, α = 0.017). Quantifications of membrane resistance and membrane potential of cortical OPCs are included in *Figure 2—figure supplement 1*. The distribution of membrane potential and membrane resistance among control and pK$_{ir}$4.1cKO cells, as well as example I-V curves of 'knockout-like' and 'wild-type-like' pK$_{ir}$4.1cKO OPCs, are included in *Figure 2—figure supplement 2*.
DOI: https://doi.org/10.7554/eLife.34829.005

The following figure supplements are available for figure 2:

**Figure supplement 1.** K$_{ir}$4.1 deletion from cortical OPCs.
DOI: https://doi.org/10.7554/eLife.34829.006

**Figure supplement 2.** EGFP$^+$ OPCs in pK$_{ir}$4.1cKO animals can be classified in two groups based on membrane currents.
DOI: https://doi.org/10.7554/eLife.34829.007

supplement 3D–F). Thus, although OPCs lacking K$_{ir}$4.1 exhibited markedly higher membrane resistance and more depolarized membrane potential, remarkably, these changes did not alter their proliferation or differentiation into oligodendrocytes.

## K$_{ir}$4.1 does not influence the membrane potential or membrane resistance of oligodendrocytes

K$_{ir}$4.1 continues to be expressed by oligodendroglia after differentiation (*Brasko et al., 2017*; *Kalsi et al., 2004*; *Neusch et al., 2001*; *Poopalasundaram et al., 2000*; *Zhang et al., 2014*), and the results above indicate that oligodendrocytes in situ are depolarized when K$_{ir}$4.1 is removed from the entire CNS (*Figure 1*), raising the possibility that cell autonomous alterations in their physiological properties induce the malformation of myelin observed in these animals. To test this hypothesis, we crossed *Kcnj10$^{fl/fl}$* mice with *Mog-iCre* mice, which express Cre recombinase specifically in mature oligodendrocytes (*Buch et al., 2005*). Unlike nK$_{ir}$4.1cKO mice, which have severe neurological symptoms and drastically shortened lifespan (*Figure 1*), these oligodendrocyte-specific Kir4.1 knockout mice (oK$_{ir}$4.1cKO) were grossly normal phenotypically and survived to adulthood.

To enable analysis of both oligodendrocytes and astrocytes, and thereby verify the specificity of K$_{ir}$4.1 deletion, oK$_{ir}$4.1cKO mice were crossed to two other lines: the RCE reporter line, to express EGFP in recombined oligodendrocytes, and a *Slc1a2-EGFP* line (*Regan et al., 2007*), in which astrocytes constitutively express EGFP. EGFP$^+$ oligodendrocytes and astrocytes from the two lines were isolated using fluorescence-activated cell sorting (FACS) from 10-week-old control and oK$_{ir}$4.1cKO mice. Levels of K$_{ir}$4.1 mRNA were quantified by reverse transcription and quantitative real-time PCR (*Figure 4—figure supplement 1*). As expected, oligodendrocytes (n = 6 mice) expressed high levels of mRNA for myelin proteins, while astrocytes (n = 5 mice) expressed higher levels of mRNA for glial fibrillary acidic protein (GFAP) (*Figure 4—figure supplement 1A–B*). K$_{ir}$4.1 mRNA levels were >10 fold lower in oK$_{ir}$4.1cKO oligodendrocytes (n = 3) than in control oligodendrocytes (n = 3) (RQ = 0.063, 95% CI = 0.035–0.112) (*Figure 4—figure supplement 1C*), while oK$_{ir}$4.1cKO (n = 3) and control (n = 2) astrocytes showed no difference in K$_{ir}$4.1 mRNA levels (RQ = 1.52, 95% CI = 0.32–7.10) (*Figure 4—figure supplement 1D*).

To assess the role of K$_{ir}$4.1 in cell-autonomously regulating the membrane properties of oligodendrocytes, EGFP$^+$ cells were targeted for whole cell recording in the corpus callosum of 5 to 6-week-old control;RCE and oK$_{ir}$4.1cKO;RCE mice. In contrast to nK$_{ir}$4.1cKO mice (*Figure 1*), no significant changes in resting membrane potential (RMP) (Control: –70 ± 1 mV (n = 17) vs. oK$_{ir}$4.1cKO: –68 ± 1 mV (n = 26), p=0.25), membrane resistance (R$_m$) (Control: 33 ± 3 MΩ vs. oK$_{ir}$4.1cKO: 40 ± 4 MΩ, p=0.22), or I-V response of EGFP$^+$ oligodendrocytes were observed following selective deletion of K$_{ir}$4.1 (*Figure 4A–B*). In an independent group of oligodendrocytes recorded in the alveus of the hippocampus, there was similarly no change in R$_m$ after K$_{ir}$4.1 deletion (Control: 27 ± 2 MΩ, n = 16 vs. oK$_{ir}$4.1cKO: 28 ± 3 MΩ, n = 18; p=0.86) (data not shown). These findings indicate that K$_{ir}$4.1 does not influence the membrane properties of mature oligodendrocytes, indicating that other K$^+$ conductances dominate at this stage.

To investigate the role of other barium-sensitive K$^+$ channels in maintaining the membrane conductance of control and oK$_{ir}$4.1cKO oligodendrocytes, membrane resistance was measured before and after application of BaCl$_2$ (100 μM). Application of BaCl$_2$ increased the membrane resistance of control and oK$_{ir}$4.1cKO oligodendrocytes to a similar degree (ΔR$_m$control: 15.2 ± 3.2 MΩ, n = 25 vs. oK$_{ir}$4.1cKO: 19.9 ± 3.5 MΩ, n = 32; p=0.33) (*Figure 4—figure supplement 2*). These results indicate

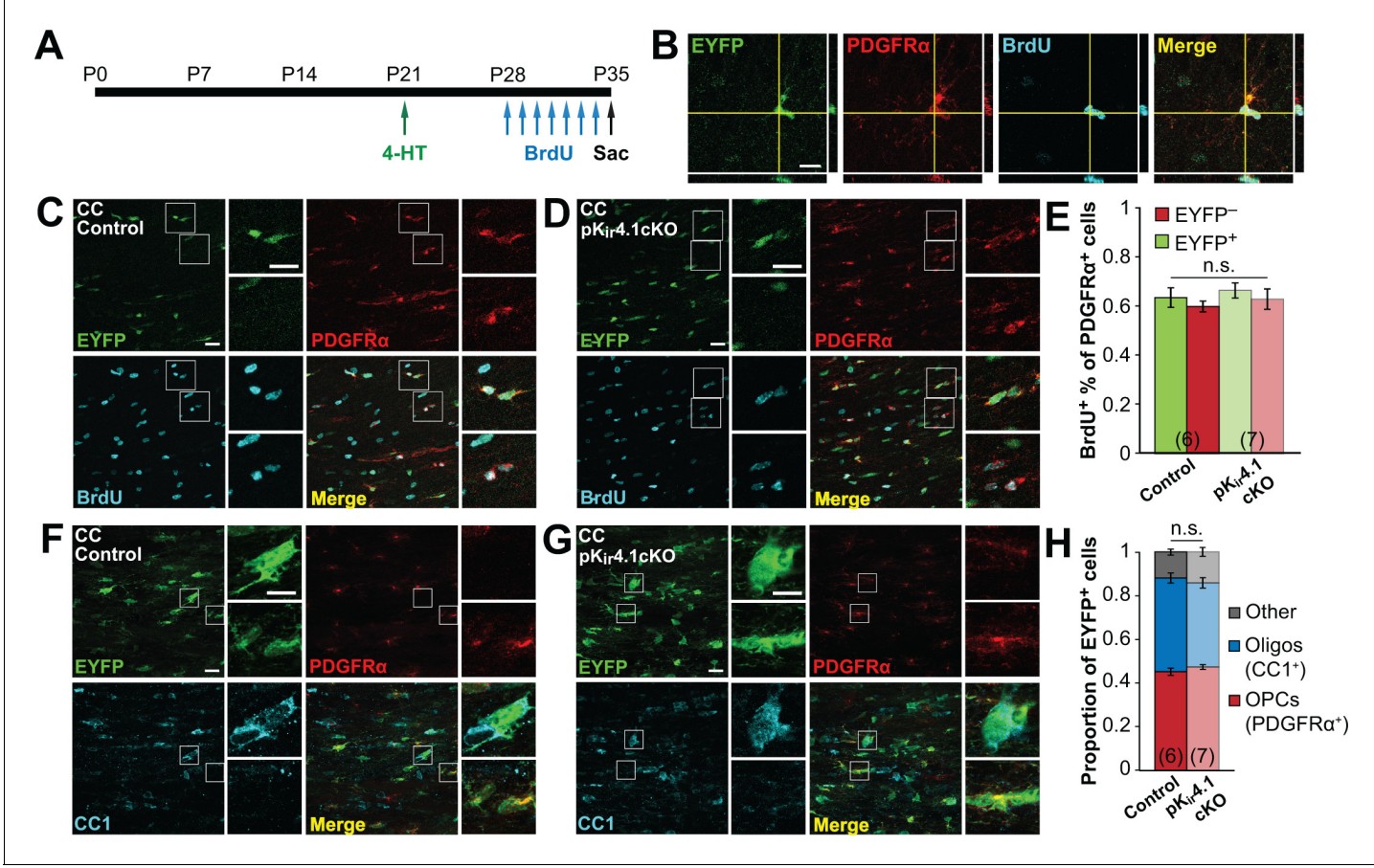

**Figure 3.** Deletion of Kir4.1 from OPCs does not affect their proliferation or differentiation. (**A**) Experimental protocol: 4-hydroxytamoxifen (2 × 1 mg) was administered to *Pdgfra-CreER;R2R-EYFP* (control) and *Pdgfra-CreER;R26R-EYFP;Kcnj10$^{fl/fl}$* (pK$_{ir}$4.1cKO) mice i.p. at P21. BrdU (2 × 50 mg/kg i.p. +1 mg/mL in drinking water) was administered daily from P28-P34, and mice were sacrificed at P35. (**B**) Immunostaining for EYFP (green), PDGFRα (red), and BrdU (cyan), showing a BrdU$^+$ OPC in the hippocampus of a control mouse. Orthogonal views show colocalization of the signals in the z plane. (**C–D**) Immunostaining for EYFP (green), PDGFRα (red), and BrdU (cyan) in the corpus callosum of a control (**C**) and pK$_{ir}$4.1cKO (**D**) mouse. Top insets show EYFP$^+$ PDGFRα$^+$ BrdU$^+$ OPCs, bottom insets show EYFP$^-$ PDGFRα$^+$ BrdU$^+$ OPCs. (**E**) Quantification of the proportions of EYFP$^+$ and EYFP$^-$ PDGFRα$^+$ cells that incorporated BrdU in the corpus callosum of control (*n* = 6) and pK$_{ir}$4.1cKO (*n* = 7) mice. There is no significant interaction between genotype and EYFP expression ($F_{interaction}$ = 9.9 × 10$^{-5}$, p=0.99) (two-way ANOVA). All pairwise comparisons between groups are not statistically significant (p>0.008; Bonferroni's test). (**F–G**), Immunostaining for EYFP (green), PDGFRα (red), and CC1 (cyan) in the corpus callosum of a control (**F**) and pK$_{ir}$4.1cKO (**G**) mouse. Top insets show EYFP$^+$ PDGFRα$^-$ CC1$^+$ oligodendrocytes, bottom insets show EYFP$^+$ PDGFRα$^+$ CC1$^-$ OPCs. (**H**) Quantification of the proportions of EYFP$^+$ cells expressing PDGFRα$^+$ (red), CC1$^+$ (blue), or neither (gray) in corpus callosum of control (*n* = 6) and pK$_{ir}$4.1cKO (*n* = 7) mice. All pairwise comparisons are not statistically significant (p>0.017; Bonferroni's test). Recombination efficiencies of the *Rosa-CAG-EGFP* and *R26R-EYFP* reporter lines are compared in *Figure 3—figure supplement 1*. OPC densities in control and pK$_{ir}$4.1cKO mice are quantified in *Figure 3—figure supplement 2*. Analysis of OPC proliferation and differentiation in hippocampus is included in *Figure 3—figure supplement 3*.

DOI: https://doi.org/10.7554/eLife.34829.008

The following figure supplements are available for figure 3:

**Figure supplement 1.** Recombination efficiencies of *ROSA26-CAG-EGFP* and *R26R-EYFP* reporter lines.
DOI: https://doi.org/10.7554/eLife.34829.009

**Figure supplement 2.** Deletion of K$_{ir}$4.1 from OPCs does not affect their density.
DOI: https://doi.org/10.7554/eLife.34829.010

**Figure supplement 3.** OPC proliferation and differentiation in hippocampus.
DOI: https://doi.org/10.7554/eLife.34829.011

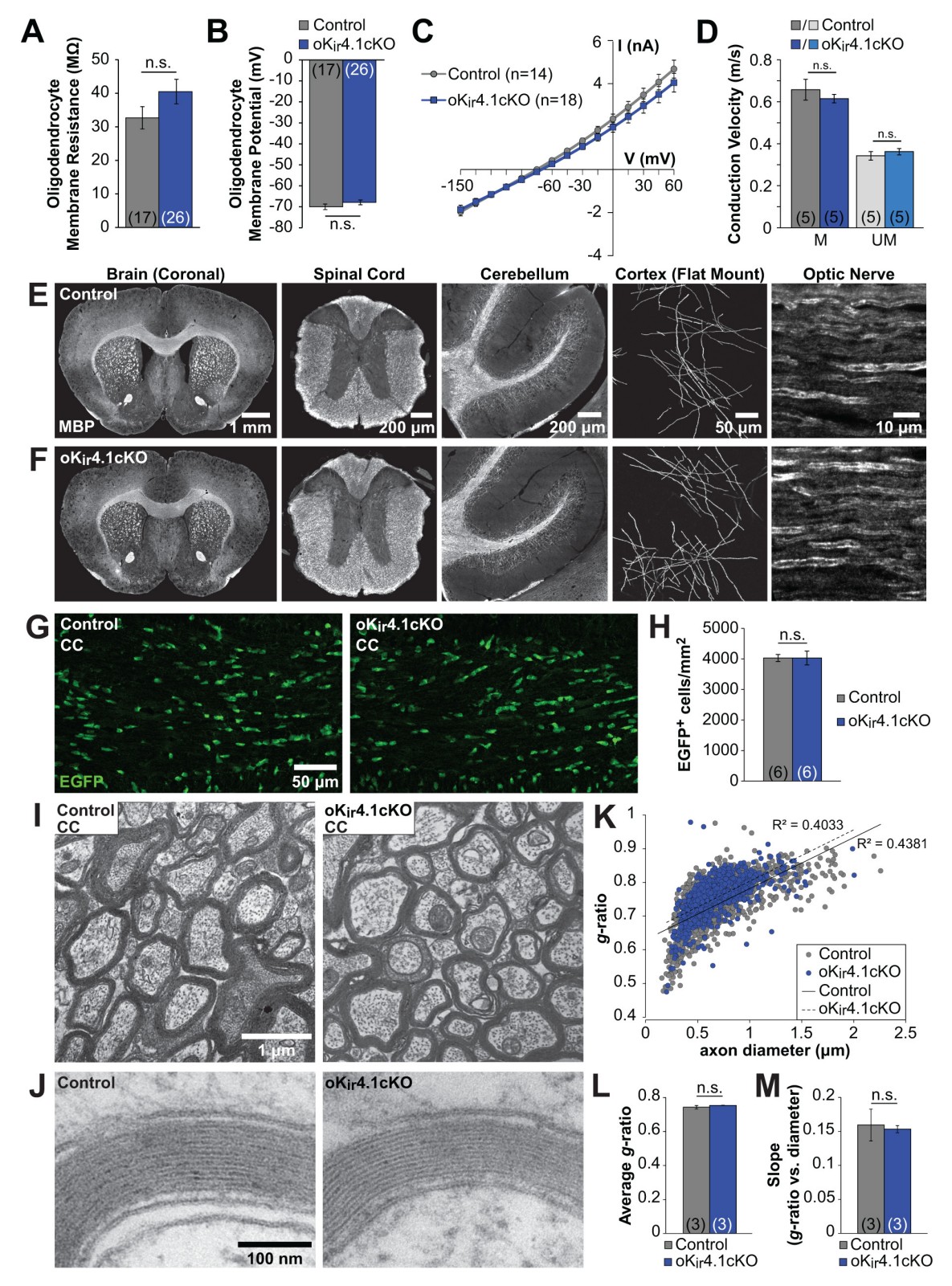

**Figure 4.** Oligodendrocyte membrane properties and myelin are preserved following selective K$_{ir}$4.1 deletion. (**A**) Membrane resistance of corpus callosum oligodendrocytes recorded in acute slices from control (*Mog-iCre;*RCE, n = 17 cells) and oK$_{ir}$4.1cKO (*Mog-iCre;*RCE;*Kcnj10$^{fl/fl}$,* n = 26 cells) mice at 5 to 6 weeks of age. No significant difference was observed in membrane resistance (p=0.22; Student's t-test). (**B**) Resting membrane potential of corpus callosum oligodendrocytes recorded in acute slices from control and oK$_{ir}$4.1cKO mice at 5 to 6 weeks of age. No significant difference was

*Figure 4 continued on next page*

*Figure 4 continued*

observed in resting membrane potential (p=0.25; Student's t-test). (C) I-V curves of control (n = 14) and oK$_{ir}$4.1cKO (n = 18) corpus callosum oligodendrocytes at 5 to 6 weeks of age. K$_{ir}$4.1 deletion did not significantly affect the current-voltage relationship. (D) Quantification of average conduction velocity of myelinated (M) and unmyelinated (UM) axons, measured via extracellular field recordings of compound action potentials in the corpus callosum of control (gray, n = 5 slices from two mice) and oK$_{ir}$4.1cKO (blue, n = 5 slices from two mice) animals. No significant differences in conduction velocity were observed (M: p=0.42, UM: p=0.37; Student's t-test). (E) Tissue sections from a 10-week-old control animal immunostained for MBP (left to right: whole brain (coronal), thoracic spinal cord (transverse), cerebellum (sagittal), layer 1 of cortex (flat mount), optic nerve (longitudinal)). (F) Tissue sections from a 10-week-old oK$_{ir}$4.1cKO animal immunostained for MBP. Same regions as in (E). (G) Immunostaining for EGFP in corpus callosum of 10-week-old control (left) and oK$_{ir}$4.1cKO (right) mice. (H) Quantification of the density of EGFP$^+$ oligodendrocytes in the corpus callosum of control (n = 6) and oK$_{ir}$4.1cKO (n = 6) mice. No significant difference was observed (p=0.997; Student's t-test). (I) Transmission electron micrographs showing myelinated axons in the corpus callosum of 10-week-old control and oK$_{ir}$4.1cKO mice. (J) Higher-magnification micrographs showing compact myelin in control and oK$_{ir}$4.1cKO mice. (K) Scatterplot of the relationship between axon diameter (X-axis) and myelin *g*-ratio (diameter of axon/diameter of axon + myelin) (Y-axis). 1103 axons from three control animals (gray) and 692 axons from 3 oK$_{ir}$4.1cKO animals (blue) were analyzed. Solid line = linear regression of control data, $R^2$ = 0.4831. Dashed line = linear regression of oK$_{ir}$4.1cKO data, $R^2$ = 0.4033. (L) Average *g*-ratio of corpus callosum axons from control (n = 3) and oK$_{ir}$4.1cKO (n = 3) animals. No significant difference was observed (p=0.38; Student's t-test). (M) Average value of the slope of the linear regression (axon diameter vs. *g*-ratio) from control (n = 3) and oK$_{ir}$4.1cKO (n = 3) animals. No significant difference was observed (p=0.81; Student's t-test). Quantitative RT-PCR demonstrating reduced K$_{ir}$4.1 expression in oK$_{ir}$4.1cKO oligodendrocytes is included in *Figure 4—figure supplement 1*. The effect of application of BaCl$_2$ to control and oK$_{ir}$4.1cKO oligodendrocytes is shown in *Figure 4—figure supplement 2*. Example traces from extracellular field recordings measuring corpus callosum axon conduction velocity are included in *Figure 4— figure supplement 3*. Immunostaining for SMI32 and GFAP in oK$_{ir}$4.1cKO mice is included in *Figure 4—figure supplement 4*.

DOI: https://doi.org/10.7554/eLife.34829.012

The following figure supplements are available for figure 4:

**Figure supplement 1.** Genetic deletion of K$_{ir}$4.1 specifically from oligodendrocytes.
DOI: https://doi.org/10.7554/eLife.34829.013

**Figure supplement 2.** Effect of barium on oligodendrocyte membrane resistance.
DOI: https://doi.org/10.7554/eLife.34829.014

**Figure supplement 3.** Example of extracellular field recordings in corpus callosum.
DOI: https://doi.org/10.7554/eLife.34829.015

**Figure supplement 4.** K$_{ir}$4.1 deletion from oligodendrocytes does not lead to axon degeneration.
DOI: https://doi.org/10.7554/eLife.34829.016

that the change in membrane resistance produced by BaCl$_2$ is due to a non-cell-autonomous effect, such as inhibition of astrocyte K$_{ir}$4.1, or that other K$^+$ channels in addition to K$_{ir}$4.1 are inhibited by BaCl$_2$.

## Specific deletion of K$_{ir}$4.1 from mature oligodendrocytes does not disrupt myelination

It has been widely hypothesized that K$_{ir}$4.1 plays an important role in myelination, as the most striking histological feature of K$_{ir}$4.1 global knockout and glia-specific knockout mice is severe spongiform vacuolation of the white matter (*Djukic et al., 2007*; *Menichella et al., 2006*; *Neusch et al., 2001*) (*Figure 1*). However, due to the broad expression of this channel, it is unclear whether this pathology is due to defects in oligodendrocytes or a consequence of astrocyte or OPC dysfunction. To determine the role of oligodendrocyte K$_{ir}$4.1 in myelination, we examined the white matter of oligodendrocyte-specific K$_{ir}$4.1 knockout mice. To assess whether axons are functionally preserved in the absence of oligodendrocyte K$_{ir}$4.1, compound action potentials (CAPs) reflecting conduction along myelinated and unmyelinated axons were recorded in the corpus callosum (*Figure 4—figure supplement 3A,B*). No difference in conduction velocity was observed for myelinated or unmyelinated axon populations between control and oK$_{ir}$4.1cKO mice (*Figure 4D*).

In contrast to the striking white matter pathology exhibited by nK$_{ir}$4.1cKO mice, the brain and spinal cord of 10-week-old oK$_{ir}$4.1cKO animals exhibited grossly normal white matter with ample myelin and no apparent vacuolation (*Figure 4E–F*). Moreover, there was no change in the density of EGFP$^+$ oligodendrocytes (*Figure 4G–H*), indicating that oligodendrocyte survival was not impaired.

To examine the structure of myelin at higher resolution, transmission electron microscopy (TEM) was performed on control and oK$_{ir}$4.1cKO mice. Electron micrographs containing cross-sections of corpus callosum from 10-week-old mice demonstrated a complete absence of vacuoles or aberrant myelin in oK$_{ir}$4.1cKO animals (*Figure 4I–J*). Furthermore, the pattern of myelination, estimated by

calculating *g*-ratios (diameter of axon/diameter of axon + myelin) (*Figure 4K*) revealed that there was no difference in *g*-ratio across all axons (Control: 0.743 ± 0.010 (*n* = 3 mice), oK$_{ir}$4.1cKO: 0.752 ± 0.001 (*n* = 3 mice), p=0.38) (*Figure 4L*), or in the relationship between axon diameter and *g*-ratio (Control: 0.16 ± 0.02 (*n* = 3 mice), oK$_{ir}$4.1cKO: 0.15 ± 0.01 (*n* = 3 mice), p=0.81) (*Figure 4M*). In addition, swollen or degenerating axons were completely absent from corpus callosum of oK$_{ir}$4.1cKO mice (*Figure 4I*) and immunostaining for SMI32 (non-phosphorylated neurofilament), which is upregulated in degenerating axons (*Lee et al., 2012*), revealed no axonal pathology in the corpus callosum of control or oK$_{ir}$4.1cKO animals (*n* = 6 for both genotypes) (*Figure 4—figure supplement 4A–B*). Brain sections were also immunostained for GFAP, as astrocytes become reactive and upregulate GFAP in response to neuronal degeneration (*Lee et al., 2012*). No difference in GFAP expression was observed between control and oK$_{ir}$4.1cKO animals (*Figure 4—figure supplement 4C–D*). These findings indicate that myelin and myelinated axons are not structurally altered as result of the selective removal of K$_{ir}$4.1channels from oligodendrocytes.

## Reduced seizure threshold in oligodendrocyte-specific K$_{ir}$4.1 knockout mice

Although oK$_{ir}$4.1cKO mice did not exhibit alterations in the structure of myelin or progressive disability, these animals died much earlier than control littermates (*Figure 5A*). K$_{ir}$4.1 is a risk factor gene for epilepsy (*Bockenhauer et al., 2009*; *Dai et al., 2015*; *Inyushin et al., 2010*; *Lenzen et al., 2005*; *Scholl et al., 2009*), raising the possibility that K$_{ir}$4.1 dysfunction in oligodendrocytes is sufficient to increase seizure susceptibility and induce death. Consistent with this hypothesis, oK$_{ir}$4.1cKO mice exhibited occasional seizures and mice that had died were often discovered with limbs extended, suggesting that they had suffered a catastrophic seizure.

To assess whether seizure susceptibility is enhanced by removal of K$_{ir}$4.1 from oligodendrocytes, we compared the response of oK$_{ir}$4.1cKO and control mice to the chemoconvulsant pentylenetetrazol (PTZ). A stark difference was observed between groups. Nearly all (11/12) oK$_{ir}$4.1cKO mice exhibited tonic-clonic seizures with full hindlimb extension after being injected with PTZ, while only 1/17 control mice exhibited this level of seizure severity (*Figure 5B*; *Figure 5—video 1*). Moreover, 8/12 oK$_{ir}$4.1cKO mice and only 1/17 control mice experienced fatal seizures, and oK$_{ir}$4.1cKO mice had a faster seizure onset (*Figure 5C–D*), indicating that deletion of this K$^{+}$ channel from oligodendrocytes enhances neuronal excitability and increases seizure susceptibility.

## K$^{+}$ buffering in white matter is impaired following K$_{ir}$4.1 deletion from oligodendrocytes

Previous studies have shown that K$_{ir}$4.1 is involved in buffering extracellular K$^{+}$ in both gray and white matter (*Bay and Butt, 2012*; *Chever et al., 2010*; *Haj-Yasein et al., 2011*; *Neusch et al., 2006*; *Sibille et al., 2014*), as extracellular K$^{+}$ levels are slower to recover following neuronal stimulation in glia-specific K$_{ir}$4.1 knockout mice (*Chever et al., 2010*). However, the relative contribution of astrocytes and oligodendrocytes to K$^{+}$ clearance, particularly in white matter where there is extensive association between axons and oligodendrocytes, has not been assessed. To investigate the role of oligodendrocyte K$_{ir}$4.1 in K$^{+}$ buffering, we recorded the response of oligodendrocytes from control and oK$_{ir}$4.1cKO mice to repetitive axonal stimulation (100 Hz for 1 s) in two white matter tracts, the corpus callosum and the alveus of the hippocampus (*Figure 6A*; *Figure 6—figure supplement 1A*).

In accordance with previous studies (*Battefeld et al., 2016*; *Yamazaki et al., 2007*; *2014*), oligodendrocytes were progressively depolarized during the stimulus train due to K$^{+}$ release from active axons and gradually returned to baseline after cessation of the stimulus (*Figure 6B–C*; *Figure 6—figure supplement 1B–C*) following the gradual decline in extracellular K$^{+}$ levels (*Battefeld et al., 2016*). Oligodendrocyte depolarization occurred more slowly in oK$_{ir}$4.1cKO mice, and the decay of membrane potential following cessation of stimulation was markedly prolonged relative to controls (*Figure 6B–D*; *Figure 6—figure supplement 1B–D*) (corpus callosum: 11.6 ± 1.6 s (*n* = 15) vs. 5.9 ± 0.7 s (*n* = 15), p=0.0036; alveus: 9.3 ± 0.6 s (*n* = 16) vs. 6.3 ± 0.4 s (*n* = 14), p=0.0017), which was also reflected in the area under the curve (*Figure 6E*; *Figure 6—figure supplement 1E*). Thus, while oligodendrocytes lacking K$_{ir}$4.1 still depolarize in response to axonal activity, presumably due

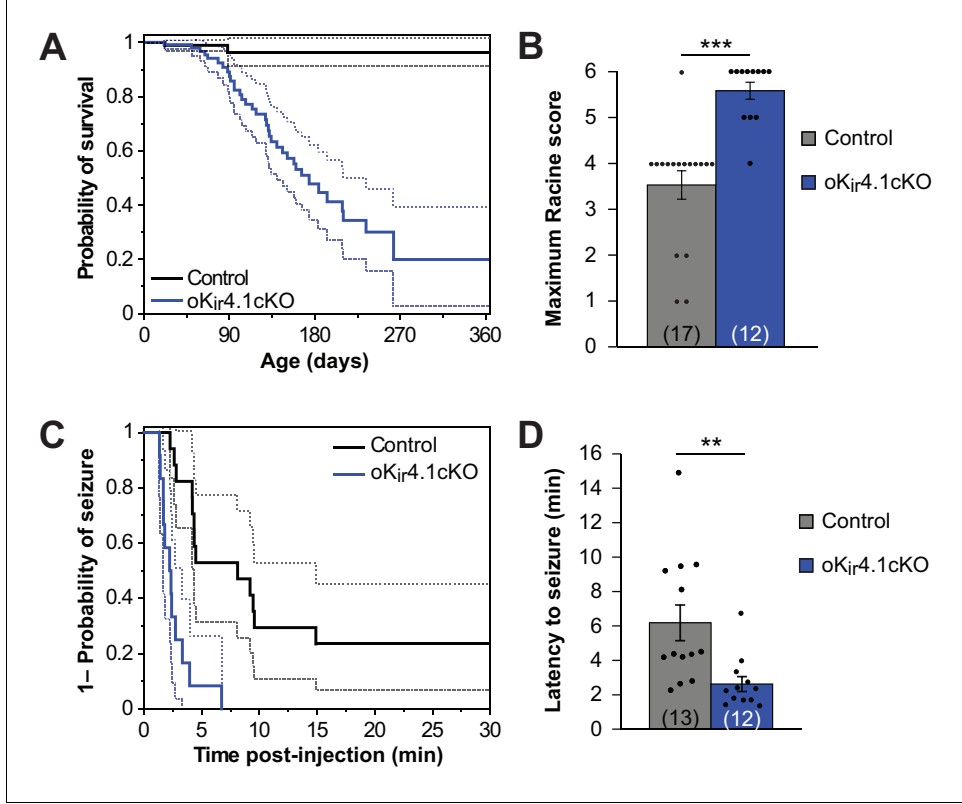

**Figure 5.** $K_{ir}4.1$ deletion from oligodendrocytes results in early mortality and reduced seizure threshold. (**A**) Kaplan-Meier curve showing the probability of survival of control (black) vs. $oK_{ir}4.1cKO$ (blue) mice from birth to one year of age. Dashed lines represent the 95% confidence interval. $oK_{ir}4.1cKO$ mice have significantly shorter survival than controls ($p=6.8 \times 10^{-7}$; log-rank test). Mice sacrificed for experimental use were censored on the date of sacrifice. Analysis is based on a total of 113 control and 129 $oK_{ir}4.1cKO$ mice. (**B**) Maximum seizure score achieved by control ($n = 17$) and $oK_{ir}4.1cKO$ ($n = 12$) mice 30 min following PTZ (40 mg/kg) injection. $oK_{ir}4.1cKO$ mice reached significantly higher seizure scores than controls ($U = -4.36$, $p=3.6 \times 10^{-5}$; Mann Whitney test). (**C**) Kaplan-Meier curve showing probability of (score $\geq3$) seizure-free survival between PTZ injection and 30 minutes post-injection. Dashed lines represent the 95% confidence interval. $oK_{ir}4.1cKO$ mice ($n = 12$) had significantly shorter seizure-free survival than controls ($n = 17$) ($p=1.2 \times 10^{-5}$; log-rank test). (**D**) Mean latency to maximal seizure among mice reaching a seizure score of 3 and above ($n = 13$ control, $n = 12$ $oK_{ir}4.1cKO$). $oK_{ir}4.1cKO$ mice had significantly shorter latency to seizure ($p=0.006$, Student's t-test).

DOI: https://doi.org/10.7554/eLife.34829.017

The following video is available for figure 5:

**Figure 5—video 1.** Control and $oK_{ir}4.1cKO$ mice undergoing PTZ-induced seizures.

DOI: https://doi.org/10.7554/eLife.34829.018

to the presence of other pathways for $K^+$ redistribution (*Rash et al., 2016*), their recovery from the stimulus train was slower, suggesting that extracellular $K^+$ clearance is slowed.

To provide an independent assessment of extracellular $K^+$ dynamics, we recorded the response of astrocytes in the corpus callosum and alveus of $oK_{ir}4.1cKO$ mice (crossed to *Slc1a2-EGFP* mice to allow astrocyte visualization) (*Figure 6F*; *Figure 6—figure supplement 1F*). Astrocyte membrane potential follows the concentration of extracellular $K^+$ closely, due to the high permeability of their membranes to $K^+$ (*Chever et al., 2010*; *Meeks and Mennerick, 2007*; *Orkand et al., 1966*). Using the same stimulation paradigm, astrocytes were observed to depolarize with a similar time course to oligodendrocytes (*Figure 6G–H*; *Figure 6—figure supplement 1G–H*). However, despite the presence of normal $K_{ir}4.1$ expression by astrocytes, a significant slowing of their decay following the end of the stimulus was observed in $oK_{ir}4.1cKO$ mice, comparable to that observed in oligodendrocytes (*Figure 6G–J*; *Figure 6—figure supplement 1G–J*). These findings suggest that oligodendrocyte $K_{ir}4.1$ channels contribute substantially to the removal of axonally released $K^+$ in white matter.

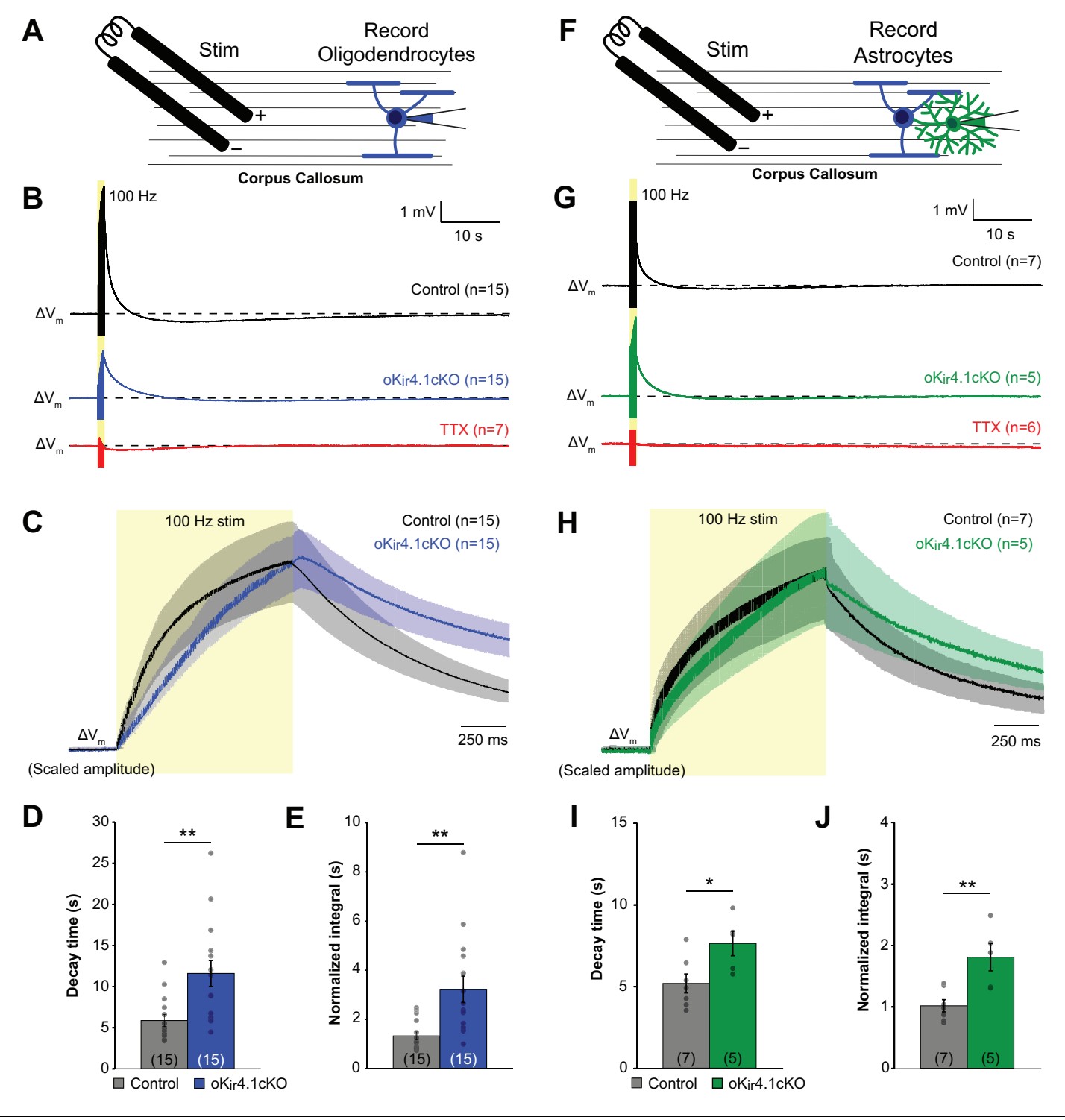

**Figure 6.** K$^+$ clearance after high-frequency stimulation of corpus callosum axons is impaired following deletion of K$_{ir}$4.1 from oligodendrocytes. (**A**) Recording set-up: EGFP$^+$ oligodendrocytes in the corpus callosum were targeted for whole cell recording in control; RCE and oK$_{ir}$4.1cKO;RCE mice. Axons were stimulated at 100 Hz for 1 s with a bipolar electrode at a distance of ~200 μm. (**B**) Membrane potential changes of oligodendrocytes recorded in current clamp mode during and after 100 Hz stimulation. Shown are averaged responses of cells from control mice (black, *n* = 15), oK$_{ir}$4.1cKO mice (blue, *n* = 15), and control mice in the presence of 1 μM TTX (red, *n* = 7). Dashed line represents baseline RMP. (**C**) Average responses of control and oK$_{ir}$4.1cKO oligodendrocytes in (**B**) shown on an expanded time scale to highlight membrane potential changes during and immediately after stimulation. Response amplitudes are scaled to the same value at the end of stimulation to facilitate comparison of response kinetics. Stimulation

*Figure 6 continued on next page*

*Figure 6 continued*

artifacts have been eliminated for clarity. (D) Quantification of the average decay time, defined as the time from the end of stimulation until the membrane potential first crosses its initial baseline value. This value is independent of response amplitude. $oK_{ir}4.1cKO$ oligodendrocytes ($n$ = 15) had significantly longer decay times than control oligodendrocytes ($n$ = 15, $p$=0.0017; Student's t-test). (E) Quantification of the average normalized integral, defined as the area under the curve from the end of stimulation until the membrane potential first crosses the baseline, divided by the peak amplitude. $oK_{ir}4.1cKO$ oligodendrocytes had significantly greater normalized integral than control oligodendrocytes ($p$=0.0022; Student's t-test). (F) EGFP$^+$ astrocytes in the corpus callosum were targeted for whole cell recording in control;*Slc1a2-EGFP* and $oK_{ir}4.1cKO$;*Slc1a2-EGFP* mice. Same protocol as in (A). (G) Membrane potential changes of astrocytes recorded in current clamp mode during and after 100 Hz stimulation. Shown are averaged responses of cells from control mice (black, $n$ = 7), $oK_{ir}4.1cKO$ mice (green, $n$ = 5), and control mice in the presence of 1 μM TTX (red, $n$ = 6). Dashed line represents baseline RMP. (H) Average responses of control and $oK_{ir}4.1cKO$ astrocytes shown in (G), on an expanded time scale. Response amplitudes were scaled to the same value at the end of stimulation, to facilitate comparison of response kinetics. Stimulation artifacts have been eliminated for clarity. (I) Quantification of the average decay time. $oK_{ir}4.1cKO$ astrocytes ($n$ = 5) had significantly longer decay times than control astrocytes ($n$ = 7, $p$=0.026; Student's t-test). (J) Quantification of the normalized integral. $oK_{ir}4.1cKO$ astrocytes had significantly greater normalized integral than control astrocytes ($p$=0.0049; Student's t-test). Similar recordings from oligodendrocytes and astrocytes in the alveus of the hippocampus are shown in *Figure 6—figure supplement 1*.

DOI: https://doi.org/10.7554/eLife.34829.019

The following figure supplement is available for figure 6:

**Figure supplement 1.** K$^+$ clearance after high-frequency stimulation of alveus is impaired following deletion of $K_{ir}4.1$ from oligodendrocytes.

DOI: https://doi.org/10.7554/eLife.34829.020

## Deletion of $K_{ir}4.1$ from oligodendrocytes affects conduction of action potentials through white matter

To assess the functional impact of oligodendrocyte-specific $K_{ir}4.1$ deletion on axonal activity in white matter, CAPs were recorded from isolated optic nerves from 10-week-old animals using suction electrodes (*Stys et al., 1991*) (*Figure 7A*). Consistent with the behavior of axons in the corpus callosum (*Figure 4D*), $K_{ir}4.1$ deletion from oligodendrocytes did not affect the mode (i.e. peak of the CAP) or median baseline conduction velocity of the nerves ($n$ = 5 nerves from three mice for each genotype) (*Figure 7B–C*).

It is established that high-frequency stimulation of the optic nerve results in an increase in extracellular K$^+$ levels and a decay of the CAP waveform (*Bay and Butt, 2012*; *Ransom et al., 2000*). In young $K_{ir}4.1$ KO optic nerves, there is a slower restoration of the extracellular K$^+$ concentration after stimulation (*Bay and Butt, 2012*), suggesting that $K_{ir}4.1$ plays an important role in K$^+$ clearance in this tissue. To assess the contribution of oligodendrocyte $K_{ir}4.1$ to K$^+$ clearance, optic nerves from control and $oK_{ir}4.1cKO$ mice were stimulated at high frequency (100 Hz), which caused the CAP waveform to progressively broaden and decrease in amplitude (*Figure 7D*). After cessation of stimulation, the CAP waveform gradually returned to its previous form (*Figure 7E*). Although $oK_{ir}4.1cKO$ nerves exhibited a similar reduction in peak amplitude during stimulation, the CAP waveform in these nerves took significantly longer to recover after stimulation ended (*Figure 7E–F*). The CAP integral, representing total activity in the nerve, and the median axon conduction latency also recovered more slowly in $oK_{ir}4.1cKO$ mice (*Figure 7—figure supplement 1*). When the same experiments were repeated using lower frequency stimulation (20 Hz), the recovery of $oK_{ir}4.1cKO$ CAPs was also delayed (*Figure 7G*, *Figure 7—figure supplement 1*). These findings suggest that oligodendrocyte-specific $K_{ir}4.1$ deletion alters action potential conduction in both physiological and pathophysiological activity regimes.

## Deletion of $K_{ir}4.1$ from oligodendrocytes results in activity-dependent motor deficits

To determine whether removal of $K_{ir}4.1$ selectively from oligodendrocytes influences normal behavior, we examined the motor function of $oK_{ir}4.1cKO$ mice. When placed in an open field chamber, there was no difference in total locomotion or in rearing behavior of $oK_{ir}4.1cKO$ mice (*Figure 8A*). However, when challenged by placement on an accelerating rotarod, $oK_{ir}4.1cKO$ mice had significantly shorter latency to fall (*Figure 8B*). Moreover, when given free access to a running wheel over multiple days, control mice increased their running activity over time, eventually running 10–15 km per 24 hr period (*Figure 8C*), while $oK_{ir}4.1cKO$ mice were unable to achieve the same performance level, and typically did not exceed 5 km per 24 hr. As a result, the cumulative distance run over 7

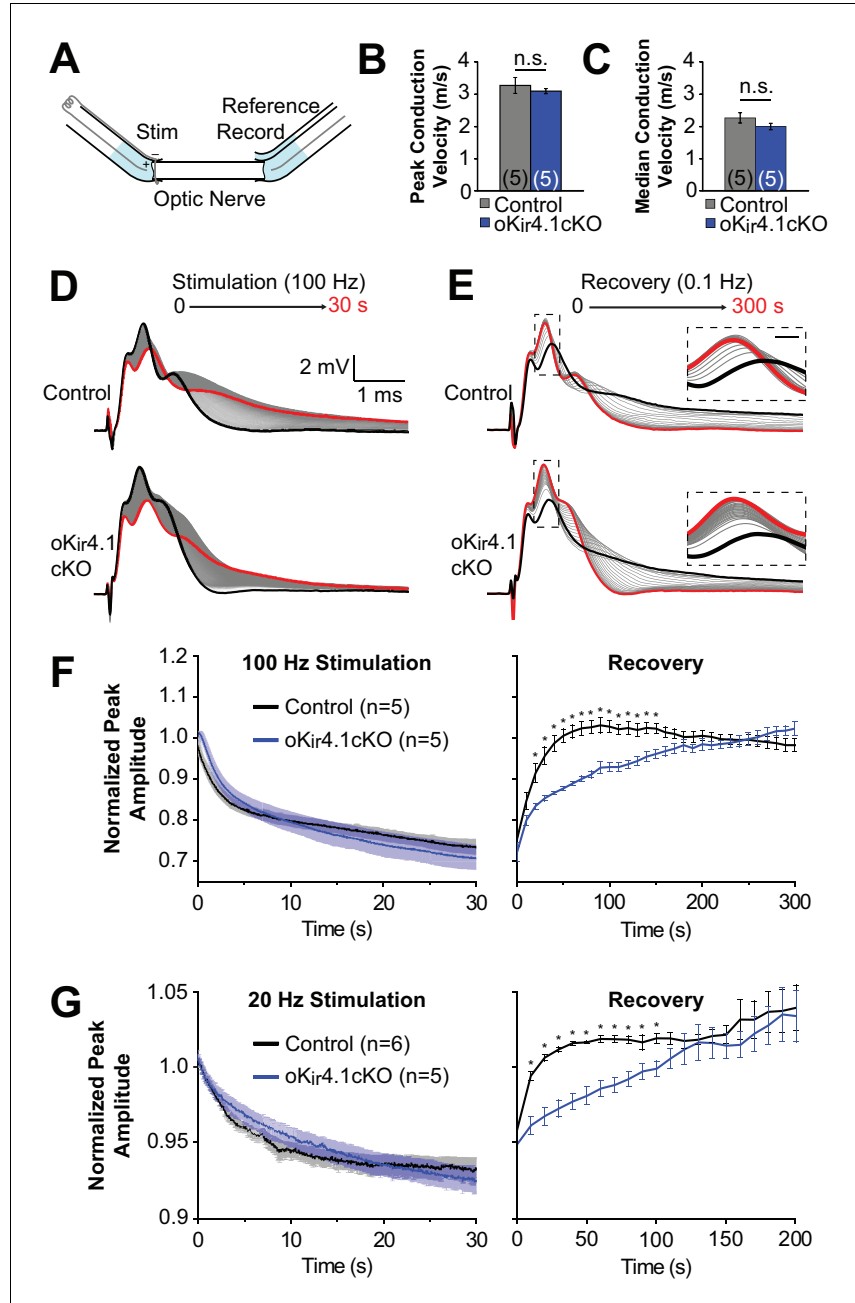

**Figure 7.** Recovery of axonal firing patterns after high frequency activity is slowed following deletion of $K_{ir}4.1$ from oligodendrocytes. (**A**) Schematic diagram of optic nerve recording set-up. Nerve is inserted into suction electrodes, stimulated at the retinal end, and recorded at the chiasmatic end. A reference electrode is placed next to the recording electrode, and the signals are subtracted to reduce the stimulus artifact. (**B**) Quantification of the peak (mode) conduction velocities of optic nerve axons from control ($n = 5$ nerves) and $oK_{ir}4.1cKO$ ($n = 5$ nerves) mice. No significant difference was observed ($p=0.51$; Student's t-test). (**C**) Quantification of the median conduction velocities of optic nerve axons from control ($n = 5$ nerves) and $oK_{ir}4.1cKO$ ($n = 5$ nerves) mice. Median was calculated over 6 ms following stimulus artifact. No significant difference was observed ($p=0.11$; Student's t-test). (**D**) CAPs recorded from a control (top) and $oK_{ir}4.1cKO$ (bottom) optic nerve during 30 s of 100 Hz stimulation. The bold black trace is the first sweep, and the bold red trace is the last sweep. (**E**) CAPs recorded at 0.1 Hz from a control (top) and $oK_{ir}4.1cKO$ (bottom) optic nerve during 5 min of recovery from stimulation. The bold black trace is the first sweep, and the bold red trace is the last sweep. Insets: higher resolution views of the CAP peaks during recovery, showing slower recovery of the $oK_{ir}4.1cKO$ nerve. (**F**) Peak CAP amplitude, as a fraction of the baseline value, of control and $oK_{ir}4.1cKO$ nerves during 100 Hz stimulation and recovery. $oK_{ir}4.1cKO$

*Figure 7 continued on next page*

*Figure 7 continued*
nerves recovered significantly more slowly than control nerves ($F_{interaction}$ = 5.14, $p$=8.5 × 10$^{-14}$; two-way ANOVA) (*=$p$ < 0.05; simple effects post-test). (G) Peak CAP amplitude, as a fraction of the baseline value, of control and oK$_{ir}$4.1cKO nerves during 20 Hz stimulation and recovery. oK$_{ir}$4.1cKO nerves recovered significantly more slowly than control nerves ($F_{interaction}$ = 1.82, $p$=0.027; two-way ANOVA) (*=$p$ < 0.05; simple effects post-test). Plots of the normalized CAP integral and median conduction latency during stimulation and recovery are included in *Figure 7—figure supplement 1*.
DOI: https://doi.org/10.7554/eLife.34829.021
The following figure supplement is available for figure 7:

**Figure supplement 1.** Recovery of CAP integral and median conduction latency are slowed following deletion of K$_{ir}$4.1 from oligodendrocytes.
DOI: https://doi.org/10.7554/eLife.34829.022

days was significantly less, as was the average speed during active running periods (*Figure 8D–E*). These results suggest that, under physiologically relevant behavioral conditions, oligodendrocyte K$_{ir}$4.1 channels play a vital role in facilitating neuronal activity.

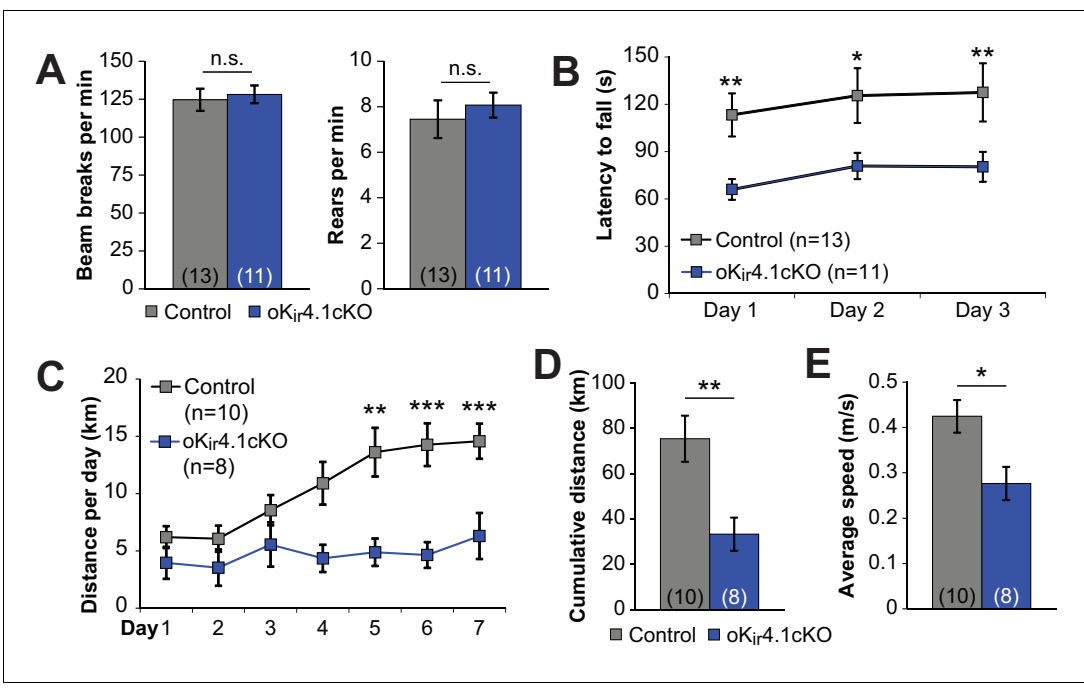

**Figure 8.** Deletion of K$_{ir}$4.1 from oligodendrocytes results in activity-dependent motor deficits. (A) Quantification of total beam breaks and rears per minute during 30 min in an open field chamber. No significant difference was observed between control ($n$ = 13) and oK$_{ir}$4.1cKO ($n$ = 11) mice in total activity ($p$=0.71; Student's t-test) or rearing ($p$=0.56; Student's t-test). (B) Latency to fall on the accelerating rotarod. Initial speed was 5 rpm, with acceleration of 1 rpm every 5 s. There was a significant relationship between genotype and performance which was independent of trial day ($F_{genotype}$ = 20.38, $p$=2.7 × 10$^{-5}$; $F_{interaction}$ = 0.007, $p$=0.99; two-way ANOVA) (*=$p$ < 0.05, **=$p$ < 0.01; simple effects post-test). (C) Distance run on a freely available running wheel per 24 hr period over 7 days of exposure. There was a significant interaction between genotype and trial day ($F_{interaction}$ = 3.09, $p$=0.008, two-way ANOVA). oKi4.1cKO mice ($n$ = 8) had significantly less daily distance than control mice ($n$ = 10) on days 5–7 (**=$p$ < 0.01, ***=$p$ < 0.001; simple effects post-test). (D) Cumulative distance run on a freely-available running wheel over 7 days of exposure. oK$_{ir}$4.1cKO mice ran significantly less distance than control mice ($p$=0.0057; Student's t-test). (E) Average running speed during intervals where running occurred. oK$_{ir}$4.1cKO mice had lower average speed than control mice ($p$=0.021; Student's t-test).
DOI: https://doi.org/10.7554/eLife.34829.023

## Discussion

Oligodendrocytes play a critical role in the mammalian CNS by forming myelin sheaths around axons that enable action potentials to be propagated rapidly with minimal energy expenditure. Although they were once assumed to be merely passive insulators, recent studies indicate that oligodendrocytes also provide metabolic support to axons in the form of lactate (*Fünfschilling et al., 2012*; *Lee et al., 2012*) and contribute to a larger glial network by coupling to astrocytes through gap junctions (*Kamasawa et al., 2005*; *Mugnaini, 1986*; *Orthmann-Murphy et al., 2007*; *Rash et al., 1997*; *2001*). Removal of oligodendrocytes by exposure to the oligotoxin cuprizone, or through genetic ablation in animal models, enhances neuronal excitability (*Hamada and Kole, 2015*) that can lead to tremor and death (*Traka et al., 2010*). Moreover, oligodendrocyte loss contributes to neurological disability in diseases such as multiple sclerosis, highlighting the importance of defining the mechanisms that control their form and function. To determine the cell-autonomous role of the inwardly rectifying $K^+$ channel $K_{ir}4.1$, which has been linked to $K^+$ clearance, neuronal hyperexcitability and myelin disruption, we selectively deleted this abundant channel from OPCs and mature oligodendrocytes in vivo. Despite high expression of this channel by oligodendroglia, genetic removal of $K_{ir}4.1$ did not alter the developmental trajectory of OPCs or the ability of oligodendrocytes to form and maintain myelin. Nevertheless, mice in which $K_{ir}4.1$ was selectively deleted from oligodendrocytes exhibited profound motor deficits that emerged with increasing activity, as well as spontaneous seizures that were often fatal. Recordings from both oligodendrocytes and astrocytes revealed that activity-induced $K^+$ clearance in white matter was impaired in these mice, and axons recovered more slowly from repetitive stimulation. Together, these findings indicate that oligodendrocyte $K_{ir}4.1$ channels are prominent contributors to $K^+$ homeostasis in white matter, and that selective loss of this channel from oligodendrocytes is sufficient to reduce motor performance, enhance neuronal hyperactivity and induce seizures.

### Myelin pathology in the absence of $K_{ir}4.1$

Global removal of $K_{ir}4.1$ leads to the formation of prominent vacuoles in myelin sheaths, failure of myelin compaction, and ultimately axonal degeneration (*Djukic et al., 2007*; *Menichella et al., 2006*; *Neusch et al., 2001*) (*Figure 1E–F*; *Figure 1—figure supplement 1*). Several explanations for this dramatic pathology have been proposed to link $K_{ir}4.1$-deficient oligodendroglia to the abnormal formation and maintenance of myelin. Loss of this channel from OPCs may compromise their viability, impair their ability to generate oligodendrocytes or cause abnormalities in oligodendrocytes that are produced. This hypothesis is supported by the observation that OPCs ('complex glia') were rarely encountered in hippocampal recordings from glial-specific $K_{ir}4.1$ knockout mice, in which $K_{ir}4.1$ was deleted from both astrocytes and oligodendroglia (*Djukic et al., 2007*). However, our studies show that OPC and oligodendrocyte densities were normal in global CNS $K_{ir}4.1$ knockout (n$K_{ir}4.1$cKO) mice (*Figure 1I–K*), despite the presence of widespread myelin vacuolization; the proliferation rate of OPCs also was not increased, as would be expected if these progenitors were mobilized to replace dying OPCs and oligodendrocytes. Moreover, when $K_{ir}4.1$ was specifically deleted from OPCs during the period of rapid oligodendrogenesis (P21), the survival, proliferation, and differentiation of these progenitors was unaffected (*Figure 3*; *Figure 3—figure supplements 2–3*), indicating that removal of $K_{ir}4.1$ specifically from OPCs does not contribute to this white matter pathology.

Oligodendrocytes in mixed glial cultures from $Kcnj10^{-/-}$ mice are significantly depolarized and fail to mature fully (*Neusch et al., 2001*). We found that they were similarly depolarized in n$K_{ir}4.1$cKO mice in situ (*Figure 1M*), raising the possibility that physiological changes in oligodendrocytes impair their ability to generate or maintain myelin sheaths. However, selective deletion of $K_{ir}4.1$ from oligodendrocytes did not alter their membrane properties, the structure of myelin (*Figure 4*), or the baseline conduction velocity of action potentials in white matter (*Figure 4D*; *Figure 7B–C*). Notably, recent studies also found that the membrane properties of satellite oligodendrocytes (oligodendrocytes located next to neuronal cell bodies) in the cerebral cortex were unaffected in *Plp1-CreER;Kcnj10^{fl/fl}* mice (*Battefeld et al., 2016*). Oligodendrocytes express a variety of $K^+$ leak channels that may compensate for the loss of $K_{ir}4.1$, including $K_{2P}$ channels and the inward-rectifying channel $K_{ir}2.1$ (*Gipson and Bordey, 2002*; *Hawkins and Butt, 2013*; *Pérez-Samartín et al., 2017*; *Stonehouse et al., 1999*; *Zhang et al., 2014*). However, the pronounced

depolarization of oligodendrocytes in $Kcnj10^{-/-}$ and $nK_{ir}4.1cKO$ mice (**Figure 1M**) indicates that this potential compensation is insufficient in the context of widespread $K_{ir}4.1$ deletion, suggesting that non-cell autonomous effects are crucial contributors to oligodendrocyte depolarization and myelin disruption.

Astrocytes in $Kcnj10^{-/-}$ mice are also severely depolarized, and their ability to remove extracellular $K^+$ is impaired (**Djukic et al., 2007**), consistent with their high expression of this channel. These dramatic changes in membrane potential could decrease their ability to provide metabolic support to surrounding cells (**Morrison et al., 2013**; **Rinholm et al., 2011**; **Sánchez-Abarca et al., 2001**), reduce the production of factors that promote oligodendrocyte development and myelination (for review, see **Kıray et al., 2016**), and disrupt ion and water balance leading to vacuolization (**Menichella et al., 2006**; **Rash, 2010**). As oligodendrocytes form gap junctions with astrocytes, they may be particularly sensitive to these changes in astrocyte physiology. In support of this hypothesis, deletion of gap junction proteins that link astrocytes and oligodendrocytes produces myelin pathology comparable to what occurs in $Kcnj10^{-/-}$ mice (**Magnotti et al., 2011**; **Menichella et al., 2003**), and heterozygote analysis indicates that there is a close genetic interaction between these connexin genes and $K_{ir}4.1$ in producing myelin vacuolization (**Menichella et al., 2006**). The presence of a glial syncytium may help to maintain normal oligodendrocyte membrane potentials when $K_{ir}4.1$ is specifically deleted from these cells and may exacerbate oligodendrocyte depolarization when $K_{ir}4.1$ is deleted globally, as astrocytes have the ability to maintain a highly stable membrane potential through extensive electric and ionic coupling between cells (**Ma et al., 2016**). Our results emphasize that both the structure and function of oligodendrocytes is critically dependent on astrocytes, and that reactive changes in astrocytes associated with reduced $K_{ir}4.1$ expression (**Hinterkeuser et al., 2000**; **Kaiser et al., 2006**; **MacFarlane and Sontheimer, 1997**; **Pivonkova et al., 2010**; **Tong et al., 2014**) may in turn precipitate pathological changes in myelin.

## Regulation of OPC behavior

OPCs are unique among macroglia in that they continue to proliferate throughout life, engaging in a homeostatic response to replace cells lost through differentiation and death (**Hughes et al., 2013**; **Robins et al., 2013**). The mechanisms that guide this remarkable behavior are not well understood, but have important consequences for myelin repair, trauma-induced gliosis and brain cancer. Membrane potential fluctuates predictably during the cell cycle in many cell types (**Blackiston et al., 2009**; **Sundelacruz et al., 2009**), and previous studies indicate that $K^+$ channels play an important role in regulating cell cycle progression in OPCs. Pharmacological inhibition of delayed rectifier $K^+$ channels in cultured OPC-like O-2A cells and OPCs in cerebellar slice cultures inhibits their proliferation and differentiation by inducing $G_1$ arrest (**Gallo et al., 1996**; **Ghiani et al., 1999**; **Knutson et al., 1997**). Subsequent studies showed that OPCs upregulate $K_v1.3$ during $G_1$ phase of the cell cycle, and blockade of this channel prevents $G_1/S$ transition (**Chittajallu et al., 2002**; **Tegla et al., 2011**), while overexpression of certain $K_v1$ isoforms promotes their proliferation, suggesting that OPC homeostasis is critically regulated by their membrane potential and potassium conductance. Although the effect of $K_{ir}4.1$ channels on OPC proliferation and lineage progression had not been evaluated, studies from developing astrocytes and glioma cells suggest that this channel is also involved in cell cycle regulation. Expression of $K_{ir}4.1$ channels during astrocyte development correlates with a negative shift in RMP, cessation of cell proliferation, and increased differentiation (**Bordey and Sontheimer, 1997**; **MacFarlane and Sontheimer, 2000**), and astrocytes that become proliferative after injury have lower $K_{ir}$ current density than non-proliferating astrocytes near the injury site (**MacFarlane and Sontheimer, 1997**). In addition, heterologous expression of $K_{ir}4.1$ in glioma cells induces hyperpolarization and $G_1$ arrest, an effect nullified if cells are treated with $BaCl_2$ to block $K_{ir}4.1$ or artificially depolarized by high $K^+$ (**Higashimori and Sontheimer, 2007**). Thus, it is remarkable that removal of $K_{ir}4.1$ from OPCs in vivo, which led to a profound shift in their RMP and increase in their membrane resistance, had no discernable effect on their proliferation, density or their ability to differentiate into oligodendrocytes (**Figure 3**; **Figure 3—figure supplements 2–3**). It is possible that OPCs overcome the depolarization induced by $K_{ir}4.1$ loss by transiently increasing $K^+$ channel expression during certain phases of the cell cycle or by reducing the activity of other channels normally recruited through depolarization, such as voltage-gated calcium channels (**Paez et al., 2007**). These findings highlight both the extreme behavioral flexibility of OPCs in vivo and the powerful drive to sustain their numbers in the adult CNS. However, our studies

also indicate that OPCs are profoundly influenced by their environment, as complete removal of $K_{ir}4.1$ from the CNS profoundly reduced their proliferation by the third postnatal week (*Figure 1K*). This phenomenon occurred in the context of widespread depolarization and dysfunction of astrocytes (*Djukic et al., 2007*) and vacuolization of myelin, suggesting that astrocytes play a crucial role in maintaining a favorable environment for, and perhaps directly facilitating, OPC proliferation and development.

Although OPC proliferation and differentiation is unchanged following $K_{ir}4.1$ deletion, it remains to be seen whether other functions of OPCs are altered in these mice. OPCs receive synaptic input from neurons (*Bergles et al., 2000*), and the strength of these inputs is likely to be altered by changes in OPC membrane potential and membrane resistance. It has been shown that OPCs are able to regulate glutamatergic neurotransmission by shedding of cleaved NG2 ectodomains (*Sakry et al., 2014*), so even subtle alterations in OPC behavior have the potential to affect synaptic communication in the brain.

## $K^+$ uptake in white matter

Axonal $K_v1$ channels that control repolarization and limit re-excitation by distal nodes are abundant underneath the myelin sheath, particularly in juxtaparanodal regions that are isolated from nodes of Ranvier by septate/septate-like junctions (*Einheber et al., 1997*; *Poliak et al., 2003*; *Rash et al., 2016*; *Wang et al., 1993*). Although a single action potential can increase the local extracellular $K^+$ concentration in unmyelinated tissue by as much as 1 mM from a resting level of 3–3.5 mM (*Baylor and Nicholls, 1969*; *Frankenhaeuser and Hodgkin, 1956*), and extracellular $K^+$ levels can rise up to 10 mM during pathological conditions such as seizures (*Moody et al., 1974*), the potential for accumulation of $K^+$ underneath myelin sheaths may be even greater due to the small peri-internodal volume and the barriers to diffusion presumed to be created by the paranodal septate junctions (*Bellinger et al., 2008*) (but see *Hirano and Dembitzer, 1969*; *Hirano and Dembitzer, 1982*; *Rash et al., 2016*). However, the mechanisms that enable clearance of extracellular $K^+$ from these spaces have not been defined. Pharmacological studies of $K^+$ buffering mechanisms suggest that $K_{ir}$ channels and the $Na^+/K^+$ ATPase pump have distinct roles in overall $K^+$ clearance. Analysis of $K^+$ dynamics arising from neuronal activity in gray matter suggest that the $Na^+/K^+$ ATPase predominates in lower activity regimes, but at higher frequencies and in situations of localized $K^+$ release, $K_{ir}4.1$ plays a more prominent role in limiting extracellular $K^+$ accumulation (*Chever et al., 2010*; *Larsen et al., 2014*; *Sibille et al., 2015*).

$K_{ir}4.1$ is concentrated within the fine processes of astrocytes that surround synapses, as well as to astrocyte endfeet that contact blood vessels (*Higashi et al., 2001*), making it ideally situated to participate in $K^+$ buffering. Indeed, astrocytes appear to dominate this process in gray matter, as deletion of $K_{ir}4.1$ from oligodendrocytes did not alter local $K^+$ accumulation in the cortex after neuronal activity (*Battefeld et al., 2016*). However, the access of astrocytes to axons in white matter is limited to nodes of Ranvier, and the processes of these fibrous astrocytes are less ramified and display less $K_{ir}4.1$ immunoreactivity than protoplasmic astrocytes in gray matter (*Higashi et al., 2001*; *Poopalasundaram et al., 2000*), which may place a greater burden on oligodendrocytes for $K^+$ redistribution. Our recordings in white matter revealed that deletion of $K_{ir}4.1$ channels from oligodendrocytes slowed both their depolarization in response to repetitive stimulation and their repolarization after the stimulus train ended (*Figure 6*; *Figure 6—figure supplement 1*), suggesting that these channels normally limit extracellular $K^+$ transients induced by neuronal activity. These changes were not restricted to the protected space beneath myelin, as the membrane potential of astrocytes, which closely follow the extracellular $K^+$ concentration (*Chever et al., 2010*; *Meeks and Mennerick, 2007*; *Orkand et al., 1966*), exhibited similarly delayed response kinetics in white matter (*Figure 6*; *Figure 6—figure supplement 1*). These findings indicate that oligodendrocyte $K_{ir}4.1$ is an important mediator of white matter extracellular $K^+$ clearance.

It is noteworthy that these changes in extracellular $K^+$ homeostasis occurred without changes in oligodendrocyte membrane conductance recorded at the cell soma. As discussed above, somatic membrane conductance may be maintained by expression of other $K^+$ channels, such as $K_{2P}$ or other $K_{ir}$ channels, which are present at baseline and may be upregulated in the setting of $K_{ir}4.1$ knockout, or by the presence of gap junction-mediated connections with astrocytes. If the conductance is dominated by gap junctions (*Orthmann-Murphy et al., 2008*), it would explain the dissociation between membrane conductance and $K^+$ buffering, as these intercellular channels do not allow direct flux of

K$^+$ to and from the extracellular space. It is also possible that K$_{ir}$4.1 is the dominant K$^+$ channel within the intermodal membrane and that these regions are electrically isolated from the soma. Loss of K$_{ir}$4.1 could then slow K$^+$ clearance from the peri-axonal space and impair the ability of myelinated axons to conduct action potentials. K$_{ir}$4.1 immunoreactivity has been observed at oligodendrocyte cell bodies and proximal processes (*Brasko et al., 2017*; *Kalsi et al., 2004*; *Poopalasundaram et al., 2000*), but it has not yet been detected within paranodal loops or internode segments of myelin. Future studies using immunogold and freeze-fracture immunolabeling in white matter may help to better define the spatial relationship between oligodendrocyte K$_{ir}$4.1 channels and axonal K$_v$ channels. In these domains, K$_{ir}$4.1 may be the primary conduit of K$^+$ uptake or may work synergistically with other modes of K$^+$ clearance that have been proposed, such as the direct movement of K$^+$ from axons into myelin through paired Kv.1:connexin 29 channels (*Rash et al., 2016*). Although removal of K$_{ir}$4.1 did not alter the resting membrane potential of oligodendrocytes, it is also possible that this manipulation indirectly altered other K$^+$ clearance mechanisms, such as uptake by the Na$^+$/K$^+$ ATPase.

## Oligodendrocyte regulation of neuronal activity

Mice in which K$_{ir}$4.1 was removed from oligodendrocytes exhibited rare, but often fatal seizures, and a dramatic reduction in PTZ-induced seizure threshold (*Figure 5*). There is extensive evidence that K$_{ir}$4.1 is a key regulator of neuronal excitability, but the involvement of oligodendrocytes in this process has not been demonstrated. Several spontaneously arising rodent models of epilepsy were found to have altered K$_{ir}$4.1 levels (*Harada et al., 2013*; *Nagao et al., 2013*), and a single-nucleotide polymorphism in the K$_{ir}$4.1 gene (*kcnj10*) was found to be responsible for differences in seizure susceptibility between C57BL6 and DBA/2 mice (*Ferraro et al., 2004*; *Inyushin et al., 2010*). In addition, polymorphisms in *KCNJ10* are risk factors for epilepsy in humans (*Buono et al., 2004*; *Dai et al., 2015*; *Guo et al., 2015*; *Lenzen et al., 2005*), and seizures are a prominent component of the human disorder SeSAME/EAST syndrome, which results from loss-of-function mutations in *KCNJ10* (*Bockenhauer et al., 2009*; *Scholl et al., 2009*). It has been assumed that the seizures are primarily caused by astrocyte abnormalities and downstream sequelae. However, our findings indicate that even in the absence of any structural changes to myelin, deletion of K$_{ir}$4.1 from oligodendrocytes alone is sufficient to dramatically lower seizure threshold and induce spontaneous seizures. These findings suggest that functional changes in oligodendrocytes, which would not be detected through analysis of myelination patterns, *g*-ratios or myelin protein expression, may contribute to epilepsy susceptibility or pathogenesis.

Our studies have focused primarily on K$^+$ clearance in white matter, which is important for the spread of activity from an initial seizure focus. The increased severity of seizures after PTZ administration in oK$_{ir}$4.1cKO mice (*Figure 5B*), as well as the rapidity with which seizures reached maximum severity after onset (see *Figure 5—video 1*), are consistent with more rapid spread through white matter tracts in these mice. Gray matter oligodendrocytes, particularly those that myelinate near the axon initial segment, that also lack K$_{ir}$4.1 in these mice may contribute to seizure initiation. However, recent studies indicate that removal of K$_{ir}$4.1 from oligodendrocytes does not alter the excitability of cortical pyramidal neurons (*Battefeld et al., 2016*), perhaps due to the local abundance of astrocyte processes. Additional studies in gray matter will help elucidate the mechanisms of seizure initiation in these mice.

Oligodendrocyte K$_{ir}$4.1 knockout mice also displayed significant impairments in motor behaviors (*Figure 8*), despite normal myelination and the absence of obvious neuronal pathology or neurodegeneration (*Figure 4—figure supplement 4*). These deficits only became apparent during high-intensity activity, consistent with an activity-dependent, rather than a neurodegenerative mechanism. Although we cannot rule out the possibility that these motor impairments result from subtle changes in neuronal health or circuit function that arise secondary to chronically altered K$^+$ homeostasis or occasional seizures, the intense, coordinated neuronal activity required for high-speed running may be particularly dependent on effective K$^+$ homeostasis. Synchronous neuronal activity is also prominent in brain states associated with diverse behaviors such as learning and sleep, raising the possibility that changes in oligodendrocyte K$_{ir}$4.1 expression due to aging or disease could impair higher order brain function.

# Materials and methods

## Key resources table

| Reagent type (species) or resource | Designation | Source or reference | Identifiers | Additional information |
|---|---|---|---|---|
| Gene (*Mus musculus*) | *Kcnj10*; K$_{ir}$4.1 | NA | OMIM: 602208 | |
| Genetic reagent (*M. musculus*) | B6.129-Kcnj10tm1Kdmc/J | K. McCarthy, UNC Chapel Hill.*Djukic et al., 2007*. PMID:17942730. | RRID:IMSR_JAX:026826 | |
| Genetic reagent (*M. musculus*) | *Mog-iCre* (knock-in) | A. Waisman, Johannes Gutenberg University. *Buch et al. (2005)*. PMID:15908920 | NA | |
| Genetic reagent (*M. musculus*) | B6.Cg-Tg(Nes-cre)1Kln/J | Jackson Laboratory | RRID:IMSR_JAX:003771 | |
| Genetic reagent (*M. musculus*) | B6N.Cg-Tg(Pdgfra-cre/ERT)467Dbe/J | Bergles Lab, Johns Hopkins University. *Kang et al. (2010)*. PMID:21092857 | RRID:IMSR_JAX:018280 | |
| Genetic reagent (*M. musculus*) | STOCK Gt(ROSA)26 Sortm1.1(CAG-EGFP)Fsh/Mmjax | G. Fishell, NYU. *Sousa et al., 2009*. PMID:19363146 | RRID:MGI:4412377 | |
| Genetic reagent (*M. musculus*) | B6.129 × 1-Gt (ROSA)26Sortm1(EYFP)Cos/J | Jackson Laboratory | RRID:IMSR_JAX:006148 | |
| Genetic reagent (*M. musculus*) | STOCK Tg(Mobp-EGFP) IN1Gsat/Mmucd | MMRRC | RRID:MMRRC_030483-UCD | |
| Genetic reagent (*M. musculus*) | *Slc1a2-EGFP* (BAC-transgenic) | J. Rothstein, Johns Hopkins University. *Regan et al. (2007)*. PMID:17581948 | NA | |
| Antibody | Anti-ASPA (rabbit polyclonal) | Genetex | Cat# GTX113389; RRID:AB_2036283 | (1:1500) |
| Antibody | Anti-BrdU (rat monoclonal) | BioRad | Cat# OBT0030G; RRID:AB_609567 | (1:500); Clone BU1/75 |
| Antibody | Anti-APC (CC1) (mouse monoclonal) | EMD Millipore (Calbiochem) | Cat# OP80; RRID:AB_2057371 | (1:50) |
| Antibody | Anti-GFAP (rabbit polyclonal) | Dako | Cat# Z0334; RRID:AB_10013382 | (1:500) |
| Antibody | Anti-GFP (chicken polyclonal) | Aves Labs | Cat# GFP-1020; RRID:AB_10000240 | (1:4000) |
| Antibody | Anti-GFP (goat polyclonal) | SICGEN | Cat# AB0020-200; RRID:AB_2333099 | (1:5000) |
| Antibody | Anti-Ki67 (rabbit polyclonal) | Abcam | Cat# Ab15580; RRID:AB_443209 | (1:1000) |
| Antibody | Anti-Kir4.1 (rabbit polyclonal) | Alomone Labs | Cat# APC-035; RRID:AB_2040120 | (1:2000) |
| Antibody | Anti-MBP (chicken polyclonal) | Aves Labs | Cat# MBP; RRID:AB_2313550 | (1:500) |
| Antibody | Anti-MBP (mouse monoclonal) | BioLegend | Cat# 808401; RRID:AB_2564741 | (1:500) |
| Antibody | Anti-NG2 (guinea pig polyclonal) | Bergles Lab, Johns Hopkins University. *Kang et al., 2013*. PMID:23542689 | NA | (1:10000) |
| Antibody | Anti-PDGFRα (rabbit polyclonal) | W. Stallcup, Burnham Institute. *Nishiyama et al., 1996*. PMID:8714520 | NA | (1:500) |

*Continued on next page*

*Continued*

| Reagent type (species) or resource | Designation | Source or reference | Identifiers | Additional information |
|---|---|---|---|---|
| Antibody | Anti-PDGFRα (rabbit polyclonal) | Cell Signaling Technology | Cat# 3174S; RRID:AB_2162345 | (1:500) |
| Antibody | Anti-Neurofilament-H (SMI32) (mouse monoclonal) | BioLegend | Cat# 801702; RRID:AB_2715852 | (1:1000) |
| Antibody | Donkey anti-chicken Alexa 488 | Jackson Immunoresearch | Cat# 703-546-155; RRID:AB_2340376 | (1:2000) |
| Antibody | Donkey anti-goat Alexa 488 | Jackson Immunoresearch | Cat# 705-546-147; RRID:AB_2340430 | (1:2000) |
| Antibody | Donkey anti-rabbit Alexa 488 | Jackson Immunoresearch | Cat# 711-546-152; RRID:AB_2340619 | (1:2000) |
| Antibody | Donkey anti-chicken Cy3 | Jackson Immunoresearch | Cat# 703-165-155; RRID:AB_2340363 | (1:2000) |
| Antibody | Donkey anti-guinea pig Cy3 | Jackson Immunoresearch | Cat# 706-166-148; RRID:AB_2340461 | (1:2000) |
| Antibody | Donkey anti-mouse Cy3 | Jackson Immunoresearch | Cat# 715-166-151; RRID:AB_2340817 | (1:2000) |
| Antibody | Donkey anti-rabbit Cy3 | Jackson Immunoresearch | Cat# 711-166-152; RRID:AB_2313568 | (1:2000) |
| Antibody | Donkey anti-chicken Alexa 647 | Jackson Immunoresearch | Cat# 703-605-155; RRID:AB_2340376 | (1:2000) |
| Antibody | Donkey anti-mouse DyLight 650 | Thermo Fisher Scientific | Cat# SA5-10169; RRID:AB_2556749 | (1:2000) |
| Antibody | Donkey anti-rabbit DyLight 650 | Thermo Fisher Scientific | Cat# SA5-10041; RRID:AB_2556621 | (1:2000) |
| Antibody | Donkey anti-rat Cy5 | Jackson Immunoresearch | Cat# 712-175-153; RRID:AB_2340672 | (1:2000) |
| Sequence-based reagent | *Cnp* primers: TTTACCCGCAAAAGCCACACA (f); CACCGTGTCCTCATCTTGAAG (r) | MGH PrimerBank | PrimerBank ID:6753476a1 | |
| Sequence-based reagent | *Mobp* primers: AGTACAGCATCTGCAAGAGCG (f); TCCTCAATCTAGTCTTCTGGCA (r) | MGH PrimerBank | PrimerBank ID:678910a1 | |
| Sequence-based reagent | *Gfap* primers: CGGAGACGCATCA CCTCTG (f); TGGAGGAGTCATTCGAGACAA (r) | MGH PrimerBank | PrimerBank ID:6678910a1 | |
| Sequence-based reagent | *Kcnj10* primers: GTCGGTCGCTAAGGTCTATTACA (f); GGCCGTCTTTCGTGAGGAC (r) | MGH PrimerBank | PrimerBank ID:34328498a1 | |
| Sequence-based reagent | *Gapdh* primers: AAGATGGTGATGGGCTTCCCG (f); TGGCAAAGTGGAGATTGTTGCC (r) | *Rhinn et al. (2008)*. PMID: 18611280 | NA | |
| Commercial assay or kit | Neural Tissue Dissociation Kit (P) | Miltenyi Biotec | Cat# 130-092-628 | |
| Commercial assay or kit | FastLane Cell cDNA Kit | Qiagen | Cat# 215011 | |
| Commercial assay or kit | QuantiTect SYBR Green PCR Kit | Qiagen | Cat# 204143 | |
| Chemical compound, drug | (Z)—4-Hydroxytamoxifen (4-HT) | Sigma-Aldrich | Cat# H7904; CAS:68392-35-8 | |
| Chemical compound, drug | 5-Bromo-2'-deoxyuridine (BrdU) | Sigma-Aldrich | Cat# B5002; CAS:59-14-3 | |
| Chemical compound, drug | Pentylenetetrazol | Sigma-Aldrich | Cat# P6500; CAS:54-95-5 | |

*Continued on next page*

*Continued*

| Reagent type (species) or resource | Designation | Source or reference | Identifiers | Additional information |
|---|---|---|---|---|
| Chemical compound, drug | Tetrodotoxin citrate | Abcam | Cat# Ab120055; CAS:18660-81-6 | |
| Software, algorithm | Adobe Illustrator CS6 | Adobe | RRID:SCR_014198 | |
| Software, algorithm | Fiji | http://fiji.sc | RRID:SCR_002285 | |
| Software, algorithm | ImageJ | https://imagej.nih.gov/ij/ | RRID:SCR_003070 | |
| Software, algorithm | Origin 8.0 | OriginLab Corp. | RRID:SCR_014212 | |
| Software, algorithm | pClamp10, pClamp9.2 | Molecular Devices | RRID:SCR_011323 | |
| Software, algorithm | Python programming language | https://www.python.org/ | RRID:SCR_008394 | |
| Software, algorithm | StepOne software | Applied Biosystems | RRID:SCR_014281 | |
| Software, algorithm | Zen Blue | Zeiss | RRID:SCR_013672 | |

All animal experiments were carried out in a strict compliance with protocols approved by the Animal Care and Use Committee at the Johns Hopkins University School of Medicine.

## Transgenic mice

The following transgenic mouse lines were used in this study: *Nes-Cre* mice (*Tronche et al., 1999*) were crossed with *Kcnj10$^{fl/fl}$* mice (*Djukic et al., 2007*) to generate a CNS-specific knockout mouse (termed nK$_{ir}$4.1cKO), in which K$_{ir}$4.1 is deleted from the entire central nervous system. *Pdgfra-CreER* mice (*Kang et al., 2010*) were crossed with *Kcnj10$^{fl/fl}$* mice to generate an inducible oligodendrocyte progenitor cell-specific knockout mouse (termed pK$_{ir}$4.1cKO), in which K$_{ir}$4.1 is deleted from oligodendrocyte progenitors upon exposure to 4-hydroxytamoxifen (4-HT). *Mog-iCre* mice (*Buch et al., 2005*) were crossed with *Kcnj10$^{fl/fl}$* mice to generate an oligodendrocyte-specific knockout mouse (termed oK$_{ir}$4.1cKO), in which K$_{ir}$4.1 is deleted from mature oligodendrocytes. In some experiments, these knockout lines were crossed to fluorescent reporter mouse lines, *Rosa-CAG-EGFP* (RCE) (*Sousa et al., 2009*) and *R26R-EYFP* mice (*Srinivas et al., 2001*), to allow expression of EGFP or EYFP, respectively, in cells that express Cre. For some experiments, knockout mice were crossed to constitutive fluorescent reporter mice, *Mobp-EGFP* mice (*Gong et al., 2003*), in which EGFP is expressed in mature oligodendrocytes, and *Slc1a2-EGFP* mice (*Regan et al., 2007*), in which EGFP is expressed in astrocytes.

## Cre activity induction and cell proliferation analysis

4-hydroxytamoxifen (4-HT, Sigma-Aldrich, St. Louis, MO) was dissolved in sunflower seed oil (Sigma) and administered to *Pdgfra-CreER;Rosa-CAG-EGFP* (RCE) and *Pdgfra-CreER;R26R-EYFP* mice (with or without *Kcnj10$^{fl/fl}$*) at P21 by two intraperitoneal (i.p.) injections of 1 mg, ≥8 hr apart. Mice were given BrdU (Sigma) in their drinking water (1 mg/mL, plus 1% sucrose to increase palatability), in addition to i.p. BrdU injections (50 mg/kg of body weight, dissolved in 0.9% saline) twice daily ≥8 hr apart for 7 days (P28-P34). Ki67 immunolabeling rather than BrdU incorporation was used for proliferation analysis nK$_{ir}$4.1cKO mice to avoid additional manipulation due to their fragile state.

## Acute brain slice preparation

P21, P35, or P35-P42 mice were deeply anesthetized with isoflurane and decapitated; brains were dissected into an ice-cold N-methyl-D-glucamine (NMDG)-based solution containing the following (in mM): 135 NMDG, 1 KCl, 1.2 KH$_2$PO$_4$, 1.5 MgCl$_2$, 0.5 CaCl$_2$, 20 choline bicarbonate, and 13 glucose (pH 7.4, 310 mOsm). Coronal forebrain slices (250 μm thick for whole cell recordings; 450 μm thick for extracellular CAP recordings) were prepared in ice-cold NMDG-based cutting solution using a vibratome (Leica VT1000S, Leica Microsystems, Wetzlar, Germany) equipped with a sapphire blade. After sectioning, slices were transferred to artificial cerebral spinal fluid (ACSF) containing the following (in mM): 119 NaCl, 2.5 KCl, 2.5 CaCl2, 1.3 MgCl$_2$, 1 NaH$_2$PO$_4$, 26.2 NaHCO$_3$, and 11 glucose (292 mOsm), maintained at 37°C for 20 min and at room temperature thereafter. Solutions were bubbled continuously with 95% O$_2$/5% CO$_2$.

## Acute slice electrophysiology and analysis

EGFP$^+$ OPCs, oligodendrocytes, and astrocytes were visualized with an upright microscope (Zeiss Axioskop 2 FS plus) equipped with differential interference contrast (DIC) optics and a filter set for GFP (Brightline, GFP-A-Basic-ZHE; Semrock, Rochester, NY). Cells were visualized using a 40x water-immersion objective (Zeiss Achroplan 40x; Carl Zeiss, Oberkochen, Germany) using DIC and GFP fluorescence signals as a guide. For whole cell recordings, the electrode solution consisted of the following (in mM): 120 CH$_3$SO$_3$H (methansulfonic acid, MeS), 10 K-EGTA, 20 HEPES, 1 MgCl$_2$, 2 Na$_2$ATP, and 0.2 Na-GTP (pH 7.3, 290 mOsm). For extracellular recordings, the electrode was filled with ACSF. Pipette resistance was 3.0–4.5 MΩ for whole cell recordings and 1.5–2.5 MΩ for extracellular recordings. Recordings were made without series resistance compensation. Unless otherwise noted, the holding potential was –80 mV. Whole cell recordings were performed at room temperature, and extracellular CAP recordings were performed at 37°C (in-line heater TC-324B, Warner Instruments, Hamden, CT). Resting membrane potential was measured in current clamp mode within 30 s of establishing the whole cell recording, and membrane resistance were calculated from a 10 mV depolarizing step in voltage clamp mode. To identify OPCs and exclude differentiated cells in *Pdgfra-CreER*;RCE mice, a 70 mV depolarizing step was applied to detect the presence of a Na$_v$ current (*De Biase et al., 2010*). The following agents were applied by addition to the superfusing ACSF: BaCl$_2$ (100 µM, Sigma); tetrodotoxin (TTX, 1 µM; Abcam, Cambridge, UK). Drugs were allowed to wash in for ≥10 min before additional recordings were made. The corpus callosum and alveus were stimulated using a bipolar stainless steel electrode (Frederick Haer Co., Bowdoin, ME; tip separation, 150 mm) connected to a constant current isolated stimulator unit (Digitimer, Ltd, Welwyn Garden City, UK) controlled by a Master-8 pulse stimulator (A.M.P.I., Jerusalem, Israel) and pClamp10 software (Molecular Devices, Sunnyvale, CA). For whole cell recordings, stimuli were 200 µA and 50 µs in duration; for extracellular recordings, stimuli varied from 100 to 300 µA. During extracellular corpus callosum CAP recordings, 1 µM TTX was applied at the end of each recording, and the resulting trace was subtracted from the CAP trace to reduce the stimulus artifact and increase clarity of the CAP.

Responses were recorded using an Axopatch 200B amplifier (Axon Instruments, Union City, CA), filtered at 1 kHz, digitized at 50 kHz (for episodic recordings) or 5 kHz (for gap-free recordings) using a Digidata 1322A digitizer (Axon Instruments), and recorded to disk using pClamp10 software (Molecular Devices). Data were analyzed offline using Clampfit (Molecular Devices) and Origin (OriginLab, Northampton, MA) software. For whole cell recordings, each recorded cell was considered a biological replicate. For extracellular CAP recordings, each brain slice was considered a biological replicate.

## Tissue fixation and immunohistochemistry

P24, P35, or P70 mice were deeply anesthetized with pentobarbital and transcardially perfused with 4% paraformaldehyde (PFA) in 0.1 M sodium phosphate buffer (pH 7.4). Brains, spinal cords, and optic nerves were isolated and post-fixed in 4% PFA for 15–20 hr at 4°C. Tissue was cryoprotected in a solution containing 30% sucrose and 0.1% sodium azide in phosphate-buffered saline (PBS). After ≥3 days of cryoprotection, brains were embedded in OCT compound (Sakura Finetek USA, Torrance, CA), frozen at −20°C, and sectioned using a cryostat (HM 550; Microm International GmbH, Walldorf, Germany). 35–40 µm thick brain and spinal cord sections were stored free-floating in PBS with 0.1% sodium azide. 14 µm thick optic nerve sections were slide-mounted and stored at −20°C until further processing.

Sections were permeabilized with 0.3% Triton X-100 in PBS for 10 min at room temperature, then transferred to 0.3% Triton X-100% and 5% normal donkey serum in PBS (blocking buffer) for 1 hr at room temperature. Sections were then incubated in primary antibody diluted in blocking buffer for 16–24 hr at 4°C. See Key Resources Table for a full list of antibodies used. For ASPA immunostaining, tissue sections were incubated in LAB solution (Polysciences, Warminster, PA) for 10 min before the blocking step. For BrdU immunostaining, sections were incubated in 2 N HCl at 37°C for 30 min, followed by neutralization with 0.1 M sodium borate buffer (pH 8.5) before the blocking step. After primary antibody incubation, sections were washed in PBS and incubated for 2–3 hr at room temperature in Alexa Fluor 488-, Cy3-, Alexa Fluor 647- or DyLight 650-conjugated secondary antibodies. Sections were slide-mounted in Aqua-Poly/Mount (Polysciences).

## Image acquisition and analysis

Images were acquired using an epifluorescence microscope (Zeiss Axio-imager M1) or a LSM 510 Meta confocal microscope (Zeiss). Confocal images represent maximum intensity projections of z-stacks, with step sizes of 0.5–3 µm. Whole brain section images were acquired as tiled arrays using an epifluorescence microscope equipped with a computer-controlled stage (Cell Observer; Zeiss or BZ-X710; Keyence, Japan), and were aligned using the microscope software (Zen software; Zeiss or BZ-X software; Keyence). Images were processed with ImageJ.

For Ki67$^+$ OPC counting and GFP$^+$ oligodendrocyte counting, two forebrain sections were analyzed for each mouse, and the results averaged. For OPC BrdU and fate tracing studies, two independent areas (one from each hemisphere) were analyzed from two brain sections for each mouse, for a total of four areas, and results were averaged. All cell countings were performed by an experimenter blinded to the animal's genotype.

## Fluorescence-activated cell sorting and RT-qPCR

P70 mice were deeply anesthetized with pentobarbital and transcardially perfused with ice-cold Hank's Balanced Salt Solution without $Ca^{2+}$ or $Mg^{2+}$ (HBSS; Gibco Laboratories, Gaithersburg, MD). Brains were isolated and single-cell suspensions were generated using the Neural Tissue Dissociation Kit (Miltenyi Biotec, Bergisch Gladbach, Germany), following the manufacturer's instructions. Myelin debris was removed using a Percoll gradient, and cells were resuspended in neurobasal media (Gibco) with 1% BSA for sorting. EGFP$^+$ cells were isolated using the MoFlo MLS high-speed cell sorter (Beckman Coulter, Brea, CA) at the Johns Hopkins School of Public Health FACS core. RNA stabilization, genomic DNA elimination, and reverse transcription were performed using the Fast-Lane Cell cDNA Kit (Qiagen, Hilden, Germany), following the manufacturer's instructions. Real time qPCR was performed using the QuantiTect SYBR Green PCR Kit (Qiagen), following the manufacturer's instructions, and using a StepOnePlus Real-Time PCR System (Applied Biosystems, Foster City, CA). Primers for the following genes were identified from the MGH PrimerBank (*Spandidos et al., 2008*; *2010*; *Wang and Seed, 2003*; *Wang et al., 2012*): *Cnp* (ID: 6753476a1), *Mobp* (ID: 6678910a1), *Gfap* (ID: 196115326 c1), and K*cnj10* (ID: 34328498a1). The following primers for *Gapdh* were also used (*Rhinn et al., 2008*): AAGATGGTGATGGGCTTCCCG (forward), TGGCAAAGTGGAGATTGTTGCC (reverse). Each reaction was performed in quintuplicate, the highest and lowest values were excluded from analysis, and the remaining three technical replicates were averaged for each animal. Data were analyzed using ExpressionSuite v1.1 software (Applied Biosystems).

## Transmission electron microscopy

P70-P100 mice were deeply anesthetized with pentobarbital and transcardially perfused with 10 mL 0.15 M Sorensen's Phosphate Buffer (SPB) containing heparin (10 IU/mL, Sigma), then with 60 mL 2.5% glutaraldehyde (Polysciences) in 0.15 M SPB (pH 7.4, 37°C then 4°C). Brains were isolated and post-fixed in 2.5% glutaraldehyde for 2–3 hr at 4°C, then transferred to 0.1% PFA in 0.15 M SPB until further processing.

Fixed tissue was dissected at 4°C, and cut into 100 µm-thick slices using a refrigerated Lancer 1000 Vibratome (Technical Products International, St. Louis, MO). Corpus callosum was further dissected and post-fixed with 1% $OsO_4$ in SPB, rinsed in dH2O, and stained *en bloc* for 16 hr with 0.5% aqueous unbuffered uranyl acetate (UAc, pH 4.5). Samples stained with aqueous UAc were dehydrated in graded ethanol series. Dehydrated samples were rinsed in 100% acetone, embedded in plastic resin (10% Epon 812, 20% Araldite 502, 70% dodecenyl acetic anhydride) with 1.25–1.5% DMP-30 (dimethyl amino phenol) added as catalyst, and polymerized at 70°C for 24 hr (*Rash et al., 1969*). Silver and pale gold sections (60–100 nm thick) were cut using a Reichert Ultracut E ultramicrotome (Leica), picked up on 200 mesh copper grids, post stained with UAc and lead citrate (*Venable and Coggeshall, 1965*), air dried, and examined by transmission electron microscopy. Thin sections were examined in a JEM1400 TEM (JEOL, Tokyo, Japan), operated at 100 kV. Digital images were obtained using an 11 MB Orius SC1000 camera (Gatan, Pleasanton, CA). All images were processed using Adobe Photoshop CS5 (Adobe Systems, San Jose, CA), with 'levels' used for contrast expansion and 'brightness/contrast' used to optimize image contrast.

## g-ratio analysis

TEM images from corpus callosum containing large numbers of myelinated axons in cross-section were selected for g-ratio analysis. g-ratio analysis was performed using ImageJ software. A threshold was applied to binarize the images and custom software (*Larson, 2018*) (available at https://github.com/valerie-ann-larson/Larson-et-al-eLife-2018; copy archived at https://github.com/elifesciences-publications/Larson-et-al-eLife-2018) was used to identify axons and calculate their cross-sectional area, from which axon diameters were calculated using the formula for the area of a circle, $A = \pi r^2$. An experimenter blinded to genotype then measured the myelin sheath thickness of each axon, and excluded any improperly detected or obliquely cut axons from analysis. At least three independent images and at least 100 axons were analyzed from each animal. To compare g-ratios between genotypes, average g-ratios and g-ratio vs. axon diameter slopes were calculated for each animal and were treated as single biological replicates.

## Seizure threshold measurement

Six- to eight-month-old mice were injected intraperitoneally with pentylenetetrazol (PTZ, 40 mg/kg; Sigma), and placed in standard mouse cages without bedding for observation. Up to four mice were placed in each of four cages for simultaneous observation, and their behavior was recorded using a digital video camera (Canon VIXIA HF R400) for 30 min after injection. Seizure activity was scored by a blinded observer using a modified Racine scale with the following scoring levels (*Mizoguchi et al., 2011*; *Schröder et al., 1993*): Stage 0 = no response, Stage 1 = ear and facial twitching, Stage 2 = convulsive waves axially through the body, Stage 3 = myoclonic jerks and rearing, Stage 4 = turning over into the lateral position, Stage 5 = generalized tonic-clonic seizures with hind-limb extension, Stage 6 = death. For seizure latency calculation, only mice achieving a seizure score of ≥Stage 3 were included. Surviving mice were euthanized at the end of the experiment.

## Optic nerve recordings and analysis

P70-P75 mice were anaesthetized with isoflurane and sacrificed by cervical dislocation followed by removal of the heart. Optic nerves were then rapidly dissected and incubated at room temperature in oxygenated ACSF for ≥30 min. Nerves were then transferred to a recording chamber superfused with oxygenated ACSF at 37°C (in-line heater TC-324B, Warner Instruments). Using gentle suction, each end of the nerve was drawn into the tip of a flared pipette electrode. The stimulating electrode (containing the retinal end of the nerve) was connected to a constant current isolated stimulator unit (Winston Electronics Co., St. Louis, MO) driven by pClamp9 software (Molecular Devices). CAPs were elicited by a 1 mA, 50 μs current pulse. The recording electrode (containing the chiasmatic end of the nerve) was connected to one input channel of a Multiclamp 700A amplifier (Axon Instruments). A second electrode, placed near the recording electrode but not in contact with the nerve, was connected to the second channel of the amplifier, and the two signals were subtracted on-line by routing through a differential amplifier (Model 440, Brownlee Precision, Santa Clara, CA), significantly reducing the stimulus artifact. Signals were filtered at 1 kHz, digitized at 100 kHz using a Digidata 1322A digitizer (Axon Instruments), and recorded to disk using pClamp9 software (Molecular Devices). Data were analyzed offline using Clampfit (Molecular Devices), Origin (OriginLab), and custom software (*Larson, 2018*) (available at https://github.com/valerie-ann-larson/Larson-et-al-eLife-2018; copy archived at https://github.com/elifesciences-publications/Larson-et-al-eLife-2018). Data collected from each optic nerve was considered a biological replicate.

## Behavioral analysis

Open field and rotarod tests were performed in the Behavior Core at the Johns Hopkins University School of Medicine. The experimenter was blinded to the genetic background of the animals during testing. Open-field test was performed using a photobeam activity system (San Diego Instruments, San Diego, CA). Mice were placed in the chamber for a single 30-min period, and movement and rearing were automatically recorded as beam breaks. The rotarod test was performed using a Rota-mex-5 rotarod (Columbus Instruments, Columbus, OH). Starting speed was 5 rpm, with acceleration of 1 rpm every 5 s. On the first test day, mice were acclimated for 20 min at 5 rpm prior to the first trial. Five trials were performed per mouse on 3 consecutive days. In order to eliminate performance outliers in an unbiased fashion (e.g. single instances of failed starts that would disproportionately

affect average time), the best and worst performances on each day were eliminated, and the remaining three trials were averaged for each mouse. Wheel running behavior was recorded using a low-profile wireless running wheel system (Med Associates, Inc., Fairfax, VT) in the home cage for 7 days. Wheel turns were wirelessly recorded in 30 s bins using Wheel Manager software (Med Associates). Data were analyzed post-hoc using custom software (Robert Cudmore, Johns Hopkins University) (*Cudmore et al., 2017*).

### Statistics

Statistical analysis was performed using Origin (OriginLab, Northampton, MA) and Excel (Microsoft Corporation, Redmond, WA) software. Sample sizes were constrained by availability of cohorts of age-matched transgenic mice and were not determined in advance. Data are expressed as mean ± SEM throughout, except for qPCR RQ data, which are expressed as a mean and 95% confidence interval, and Kaplan-Meier survival curves, in which the dashed lines represent the 95% confidence interval. For maximal seizure score comparison, the non-parametric Mann Whitney test was used, as data were not normally distributed. To compare Kaplan-Meier survival curves, the log-rank test was used. For multiple comparisons, one-way or two-way ANOVA with Bonferroni's post-test was performed. For optic nerve CAP recordings, rotarod performance, and running wheel performance, two-way ANOVA was performed with simple effects post-test at each time point. All other comparisons were performed using the unpaired student's t-test, with results considered significant at $p < 0.05$.

## Acknowledgements

We thank Dr. M Pucak and N Ye for technical assistance, Dr. R Cudmore for assistance with wheel running experiments and analysis, Dr. P Schrager for technical advice with the optic nerve recordings, Dr. K McCarthy (UNC) for providing *Kcnj10*$^{fl/fl}$ mice, Dr. J Rothstein for providing *Slc1a2-EGFP* mice, Dr. D Larson for technical assistance with analysis of myelin *g*-ratios and optic nerve recordings, T Shelly for machining expertise, T Faust for help with the PTZ experiments, H Zhang for technical assistance with flow cytometry, and members of the Bergles laboratory for discussions. Funding was provided by grants from the NIH (NS051509, NS050274, NS080153), Target ALS, a Collaborative MS Research Center grant, and the Dr. Miriam and Sheldon G Adelson Medical Research Foundation to DEB.

## Additional information

### Competing interests

Dwight E Bergles: Reviewing editor, *eLife*. The other authors declare that no competing interests exist.

### Funding

| Funder | Grant reference number | Author |
|---|---|---|
| National Institutes of Health | NS080153 | John E Rash |
| Dr. Miriam and Sheldon G. Adelson Medical Research Foundation | | Dwight E Bergles |
| Target ALS | | Dwight E Bergles |
| National Institutes of Health | NS050274 | Dwight E Bergles |
| National Institutes of Health | NS051509 | Dwight E Bergles |

The funders had no role in study design, data collection and interpretation, or the decision to submit the work for publication.

## Author contributions
Valerie A Larson, Conceptualization, Data curation, Formal analysis, Validation, Investigation, Visualization, Methodology, Writing—original draft, Writing—review and editing; Yevgeniya Mironova, Formal analysis, Investigation, Writing—review and editing; Kimberly G Vanderpool, Formal analysis, Investigation; Ari Waisman, Resources, Writing—review and editing; John E Rash, Formal analysis, Funding acquisition, Methodology, Writing—review and editing; Amit Agarwal, Conceptualization, Formal analysis, Supervision, Investigation, Methodology, Writing—review and editing; Dwight E Bergles, Conceptualization, Resources, Supervision, Methodology, Writing—original draft, Project administration, Writing—review and editing

## Author ORCIDs
Valerie A Larson http://orcid.org/0000-0003-0778-0305
Amit Agarwal https://orcid.org/0000-0001-7948-4498
Dwight E Bergles http://orcid.org/0000-0002-7133-7378

## Ethics
Animal experimentation: This study was performed in strict accordance with the recommendations in the Guide for the Care and Use of Laboratory Animals of the National Institutes of Health. All of the animals were handled according to approved institutional animal care and use committee (IACUC) protocols (#MO14M310) of the Johns Hopkins University.

## Decision letter and Author response
Decision letter https://doi.org/10.7554/eLife.34829.026
Author response https://doi.org/10.7554/eLife.34829.027

# Additional files

## Supplementary files
• Transparent reporting form
DOI: https://doi.org/10.7554/eLife.34829.024

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
