## [Decision Letter]

Thank you for submitting your article "Oligodendrocyte control of potassium uptake and neuronal hyperexcitability in white matter" for consideration by *eLife*. Your article has been reviewed by three peer reviewers, and the evaluation has been overseen by Gary Westbrook as the Senior Editor. The following individuals involved in review of your submission have agreed to reveal their identity: Harald Sontheimer (Reviewer #1); David A Lyons (Reviewer #3). The reviewers have discussed the reviews with one another and the Senior Editor has drafted this decision to help you prepare a revised submission.

Summary:

As you will see from the general assessments of the three reviewers below, there was general agreement on the importance of the experiments and with the presentation and discussion of the work. The general assessments have been included in their original form for your information. There were several points that we think can be addressed with additional discussion in the text of the manuscript, and perhaps with additional data if it is available. There were also a number of minor points that we think would improve the clarity of the manuscript. We hope you will be able to submit the revised version within two months.

Essential Revisions:

1) Apparent contradiction between decreased K^+^ buffering and lack of effect on the input resistance and RP of OLs. It is not obvious how there is a change in K^+^ buffering by OLs if there is no change in resting potential or input resistance. In Figure 4B that there is no change of OL resting potential (measured at the soma) when Kir4.1 is deleted. This must mean one of the following:

i) Kir4.1 contributes a negligible conductance to the cell membrane; or

ii) the resting potential (and apparent conductance) is maintained by gap junctional coupling to astrocytes; or

iii) Kir4.1 is located in parts of the cell that are electrically distant from the soma, such as the internodal membrane; or

iv) in the KO there is a corresponding upregulation of another K^+^ conductance; or

v) in the KO there is a downregulation of a channel with a depolarized reversal potential.

Options (i)-(iv) would lead to little or no change of input resistance in the KO, while (v) would lead to an increase of resistance. Experimentally there was no significant change in resistance (though Figure 4A indicates a small increase that might become significant if more cells were studied).

However, there is apparently a significant decrease in K^+^ buffering by the cells in the KO (Figure 6), which must imply that there is less K^+^ permeability/uptake in the OL membrane, suggesting either that Kir4.1 is in fact a significant fraction of the membrane conductance or that option (ii), (iii) or (v) holds. The simplest interpretation of the data would be option (iii), but is the internodal membrane really electrically distant from the soma?

The authors gloss over the apparent contradiction between the lack of change of input resistance and the decrease in K^+^ buffering which is supposed to explain the tendency to seizures. Although further experiments to resolve this may be beyond the scope of this paper, the issues above need to be discussed frankly, along with the following points which may contribute to the explanation.

a) Is there a decrease of Na/K pump activity in the KO, since this also contributes to K^+^ buffering?

b) Is the alteration of buffering that induces seizures in all OLs, just in grey matter OLs or just in white matter OLs (the title of the paper implies white matter, but the first OL near the spike-initiation zone of the axon in the grey matter may be the key locus for epileptogenesis)?

c) Is some quantitative modeling possible of how much the gK in the internode would have to change to explain the change of K^+^ clearance seen in Figure 6, and whether this could be reconciled with the insignificant Rm change by proposing that the Kir4.1 channels are located in the (electrically distant?) internodal membrane?

d) Barium is used to mimic the KO; does this increase Rm (studying the resistance in the same cells without and with Kir4.1 blocked would increase the statistical power for detecting an Rm increase, although effects of astrocyte channels might be a confound if OL-astrocyte gap junctional coupling is strong)?

2) Change of excitability. The authors state that deleting Kir4.1 from OLs increases neuronal excitability. It's true that the animals are prone to seizures and K^+^ clearance is slower, but excitability would normally be defined as the current needed to excite an action potential in a neuron. No real measurements of excitability have been provided. If the authors want to talk about excitability, it should be useful for the authors to compare the excitability of neurons (in WT vs KO) in e.g. hippocampal or cortical slices by injecting current steps and measuring the current needed to evoke an action potential. One might expect this to be unchanged in the KO. This could then be repeated after axonal stimulation at an elevated frequency (when presumably the larger longer [K^+^]o rise would bring out a difference in action potential generation between the WT and KO). That scenario would suggest that the intrinsic excitability of neurons is unchanged, but they get more excited because K^+^ clearance is slowed. It would be good to clarify this point with experiments or discussion.

3) Title. One of the reviewers questioned the emphasis on "white matter." Is there anything to suggest that oligodendrocyes (and indeed myelinated axons) in the grey matter are not affected in the mutant mice analysed in this study. If the authors agree with this point, the title of the manuscript could be revised accordingly.

4) Statistics. Should the statistical analyses relating to Figure 2I first be a one way ANOVA followed by an appropriate post-hoc test, and for Figure 7F,G and Figure 8B,C, should the appropriate test be a TWO-way ANOVA, again followed by appropriate post-hoc texts.

Reviewer #1:

This paper evaluated the consequence of cell type specific genetic elimination of Kir4.1 from oligodendrocyte progenitor cells (OPC) in vivo. The authors used a conditional inducible knockout strategy and show selectively removal of Kir4.1 from OPCs and the oligodendrocytes developing from them. The authors find that although these cells are severely depolarized they nevertheless form normal oligodendrocytes and myelination is undisrupted. This is in difference to the non-inducible knockout performed by other prior. Interestingly, however, some of the same animals show spontaneous seizure and essentially all animals have a reduced seizure threshold. This appears to be due to impairments in K^+^ homeostasis as clearance of K^+^ is slowed in white matter. Hence the paper describes an important role for Kir4.1 in K^+^ homeostasis in white matter, which, when deficient, may cause seizures. This is a very thorough and comprehensive manuscript that advances our understanding on Kir4.1 function in white matter significantly. Elegant use of transgenic technology allowed the authors to interrogate their hypothesis without being impeded by developmental changes or compensation. The authors use a comprehensive arsenal of physiological testing ranging from patch-clamp electrophysiology to nerve conductance studies and eventually behavior. In fact, I couldn't think of any meaningful additional studies.

Reviewer #2:

This is an interesting paper that reports surprising observations on the effect of knocking out the Kir4.1 channel in oligodendrocyte-lineage cells. Previous work had shown that a global KO (or clinically-occurring mutations) produced white matter vacuolization (an observation reproduced here with brain-specific KO of the channel), which was attributed to a loss of K^+^ buffering by OLs. Surprisingly, in this paper, when this channel is deleted only in OLs, this vacuolization is not observed, indeed the effects on the properties of the white matter and OLs are apparently rather minor. Thus, many of the effects of global KO of, or mutations in, Kir4.1 may reflect an alteration of the properties of astrocytes (which also express the channel). Nevertheless, deletion of Kir4.1 in OLs alone is sufficient to induce epilepsy in the animals, which is attributed to decreased K^+^ clearance by OLs in the white matter.

Reviewer #3:

The authors combine genetic targeting of the inward rectifying potassium channel Kir4.1 with electrophysiology and rigorous marker analyses to draw the conclusion that Kir4.1 has an important function in oligodendrocytes to regulate potassium ion homeostasis and maintain normal conduction and function. The implications of the study are not only of a novel and essential role of oligodendrocytes in regulating circuit health and function, but also that astrocyte-oligodendrocyte communication is essential for maintaining myelin health. The experiments are in general executed to the highest standard and the general conclusion of sufficiently broad interest to the readership of *eLife*. Indeed, perhaps the authors could discuss the implications of their work even a little further in the context of axon-oligodendrocyte interactions that dynamically tune nervous system function. The authors have been admirably circumspect in discussing the implications of their results but could allow themselves a little more leeway.

---

## [Author Response]

Essential Revisions:1) Apparent contradiction between decreased K^+^ buffering and lack of effect on the input resistance and RP of OLs. It is not obvious how there is a change in K^+^ buffering by OLs if there is no change in resting potential or input resistance. In Figure 4B that there is no change of OL resting potential (measured at the soma) when Kir4.1 is deleted. This must mean one of the following:i) Kir4.1 contributes a negligible conductance to the cell membrane; orii) the resting potential (and apparent conductance) is maintained by gap junctional coupling to astrocytes; oriii) Kir4.1 is located in parts of the cell that are electrically distant from the soma, such as the internodal membrane; oriv) in the KO there is a corresponding upregulation of another K^+^ conductance; orv) in the KO there is a downregulation of a channel with a depolarized reversal potential.Options (i)-(iv) would lead to little or no change of input resistance in the KO, while (v) would lead to an increase of resistance. Experimentally there was no significant change in resistance (though Figure 4A indicates a small increase that might become significant if more cells were studied).

We thank the reviewer for detailing possible explanations for the reduction in K^+^ buffering by oligodendrocytes after K_ir_4.1 deletion, despite the apparent lack of change in input resistance. Although we are not able to provide a definitive explanation, based on currently available methodologies, we provide data from additional experiments to narrow down the possibilities and have added these results and additional discussion to the manuscript to highlight the most probable scenarios.

Our results indicate that (i) is true, because input resistance is not significantly altered by K_ir_4.1 deletion. We now include additional data in the text (subsection “K^+^ uptake in white matter”) for input resistance analysis from oligodendrocytes in the alveus. As in the corpus callosum, there was no significant difference in input resistance between control and K_ir_4.1 KO oligodendrocytes, indicating that K_ir_4.1 does not dominate the somatic conductance. As noted by the reviewer, the primary conductance could be mediated by gap junctions (Orthmann-Murphy et al., 2008; Wasseff and Scherer, 2011), which had been shown to contribute significantly to membrane conductance in astrocytes (Blomstrand et al., 2004; Wallraff et al., 2006), or other K^+^ channels. The latter conclusion is supported by the lack of change in resting potential in these cells after K_ir_4.1 deletion, indicating that K^+^ selective channels still dominate the membrane conductance. As noted in the Discussion section, previous work has demonstrated that oligodendrocytes express K_2P_ channels and the inward-rectifying channel K_ir_2.1 (Gipson and Bordey, 2002; Hawkins and Butt, 2013; Pérez-Samartín et al., 2017; Stonehouse et al., 1999; Zhang et al., 2014). If the former were true, then astrocytes could help clamp the membrane potential of oligodendrocytes at E_K_, despite the loss of K^+^ conductance. As the reviewer notes in (iii), an important caveat is that we do not know the length constant of an oligodendrocyte. It is possible that K_ir_4.1 dominates the membrane conductance within internode segments but is only a minor component at the soma. At present, it is not possible to assess membrane resistance within these processes. As suggested in (iv), we cannot exclude the possibility that there has been a commensurate increase in another K^+^ channel that compensates for K_ir_4.1. However, if this were true, then we would not expect to observe a change in K^+^ buffering by these cells. The scenario outlined in point (v) to maintain the same resting membrane potential, in which K_ir_4.1 deletion leads to downregulation of another channel with a more depolarized reversal potential (i.e. chloride or mixed cation) is also unlikely, as this net decrease in channel expression should be accompanied by an increase in membrane resistance, which is not observed. Thus, our results are most consistent with concepts i-iii above. We have added additional data regarding membrane resistance measurements from OLs (subsection “K^+^ uptake in white matter”) and text to the Discussion section to support this conclusion.

However, there is apparently a significant decrease in K^+^ buffering by the cells in the KO (Figure 6), which must imply that there is less K^+^ permeability/uptake in the OL membrane, suggesting either that Kir4.1 is in fact a significant fraction of the membrane conductance or that option (ii), (iii) or (v) holds. The simplest interpretation of the data would be option (iii), but is the internodal membrane really electrically distant from the soma?

As noted above, our results favor option (iii). If the remaining conductance in K_ir_4.1 KO oligodendrocytes is mediated primarily by gap junctions with astrocytes (or other oligos), these intercellular channels would not contribute to potassium uptake, leading to a dissociation between membrane conductance and uptake. Regarding options (i) and (iv), other K^+^ channels may be sufficient to maintain baseline conductance but may not pass inward current as efficiently as K_ir_ channels, leading to impaired uptake.

The authors gloss over the apparent contradiction between the lack of change of input resistance and the decrease in K^+^ buffering which is supposed to explain the tendency to seizures. Although further experiments to resolve this may be beyond the scope of this paper, the issues above need to be discussed frankly, along with the following points which may contribute to the explanation.a) Is there a decrease of Na/K pump activity in the KO, since this also contributes to K^+^ buffering?b) Is the alteration of buffering that induces seizures in all OLs, just in grey matter OLs or just in white matter OLs (the title of the paper implies white matter, but the first OL near the spike-initiation zone of the axon in the grey matter may be the key locus for epileptogenesis)?

We thank the reviewer for raising these important issues. As we have not selectively deleted K_ir_4.1 from white matter oligodendrocytes, it is possible that alterations in gray matter oligodendrocytes also contribute to seizure susceptibility in these mice. Our conclusion that this effect is primarily mediated by changes in white matter is based on evidence that extracellular K^+^ transients are prolonged in white matter (see Figure 6), and recent findings that oligodendroglial deletion of K_ir_4.1 did not alter the excitability of cortical neurons (Battefeld et al., 2016). Although oligodendrocytes have a preferred location near the AIS, this region is also surrounded by astrocytes that maintain an even higher K_ir_4.1 density, which may overcome oligodendrocyte deficits in K^+^ buffering. Nevertheless, as suggested by the reviewer, it is possible that deficits in K^+^ buffering in both gray and white matter contribute to the increase in seizure susceptibility, with changes in K^+^ buffering around the AIS supporting seizure initiation and effects in white matter promoting greater spread of seizures, leading to more severe and more generalized events (see Figure 5). It is possible that loss of K_ir_4.1 could lead to a change in Na/K pump expression, although this effect has not been reported following deletion of K_ir_4.1 in either astrocytes or oligodendrocytes. The lack of change in resting membrane potential, the ability of the cells to restore their membrane potential after tetanic stimulation (Figure 6) and both the normal survival and structure of oligodendrocytes when K_ir_4.1 was deleted suggests that Na/K pump activity, which is required to establish and maintain the membrane potential, has not been substantially reduced. We have added additional discussion about these issues to the text (Discussion section).

c) Is some quantitative modeling possible of how much the gK in the internode would have to change to explain the change of K^+^ clearance seen in Figure 6, and whether this could be reconciled with the insignificant Rm change by proposing that the Kir4.1 channels are located in the (electrically distant?) internodal membrane?

Modeling K^+^ changes in these compartments could help to define constraints on K^+^ buffering in white matter; however, we feel that this effort is beyond the scope of this study. The many undefined variables, such as the types and densities of channels located in these regions, pump densities, the density and location of gap junctions, and extracellular resistances would make it difficult to constrain the model without considerable further experimentation.

d) Barium is used to mimic the KO; does this increase Rm (studying the resistance in the same cells without and with Kir4.1 blocked would increase the statistical power for detecting an Rm increase, although effects of astrocyte channels might be a confound if OL-astrocyte gap junctional coupling is strong)?

We used barium to explore whether its effects can be ascribed exclusively to K_ir_4.1 channel inhibition and to relate our findings to the many studies in which barium has been used to block K_ir_4.1. As shown in new Figure 4—figure supplement 2, barium increases the R_m_ of oligodendrocytes, but this effect was the same in control and the cKO animals. indicating that it is either a non-cell-autonomous effect, likely due to changes in astrocytes, or that barium also inhibits other K^+^ channels in oligodendrocytes.

2) Change of excitability. The authors state that deleting Kir4.1 from OLs increases neuronal excitability. It's true that the animals are prone to seizures and K^+^ clearance is slower, but excitability would normally be defined as the current needed to excite an action potential in a neuron. No real measurements of excitability have been provided. If the authors want to talk about excitability, it should be useful for the authors to compare the excitability of neurons (in WT vs KO) in e.g. hippocampal or cortical slices by injecting current steps and measuring the current needed to evoke an action potential. One might expect this to be unchanged in the KO. This could then be repeated after axonal stimulation at an elevated frequency (when presumably the larger longer [K^+^]o rise would bring out a difference in action potential generation between the WT and KO). That scenario would suggest that the intrinsic excitability of neurons is unchanged, but they get more excited because K^+^ clearance is slowed. It would be good to clarify this point with experiments or discussion.

This is an excellent point. While other studies suggest that neuronal excitability is increased when K_ir_4.1 is deleted from astrocytes, our studies have focused on K^+^ buffering. Our findings in optic nerve, showing a prolonged CAP recovery time in oK_ir_4.1cKO mice (Figure 7) are consistent with activity-dependent changes in axonal excitability. We spent considerable time attempting to replicate the findings of Yamazaki et al., who showed that oligodendrocyte depolarization altered axonal conduction using antidromic spikes (Yamazaki et al., 2007). However, we were unable to reliably elicit antidromic spikes with tetanic stimulation (most often this led to direct excitation of the AIS). Although we are working to develop another means to address this possibility, we feel that this is currently outside the scope of this study, due to the technical challenges associated with this approach. Accordingly, we have revised statements related to excitability changes resulting from oligodendrocyte K_ir_4.1 deletion.

3) Title. One of the reviewers questioned the emphasis on "white matter." Is there anything to suggest that oligodendrocyes (and indeed myelinated axons) in the grey matter are not affected in the mutant mice analysed in this study. If the authors agree with this point, the title of the manuscript could be revised accordingly.

As noted above (point #1), it is possible that removal of K_ir_4.1 from gray matter oligodendrocytes contributes the spontaneous seizures and reduced seizure threshold in these mice. However, we have only studied white matter oligodendrocytes directly here. We posit that oligodendrocytes play a greater role in potassium clearance in white matter, as they are more numerous, and astrocytes do not have access to neuronal potassium channels in this region. We have modified the title to reflect this consideration.

4) Statistics. Should the statistical analyses relating to Figure 2I first be a one way ANOVA followed by an appropriate post-hoc test, and for Figure 7F,G and Figure 8B,C, should the appropriate test be a TWO-way ANOVA, again followed by appropriate post-hoc texts.

This has been revised as suggested.